# The functional overlap between respiration and global signal and its behavioral relevance
Jing Yuan [1], Yuejia Luo [1,2,3] ✉ & Jianfeng Zhang [4] ✉

Resting-state fMRI studies encounter the challenge of interpreting fluctuations in the global signal (GS). The GS has been linked to arousal, vigilance states, cognition, and psychiatric disorders, suggesting its functional relevance. However, GS also partially arises from physiological factors, particularly respiration. In this study, we investigate whether respiration and GS exhibit functional topographic overlap in the brain and its impact on behavior. Using resting-state fMRI data from the Human Connectome Project (*N* = 770), we find strong spatial consistency between GS and respiration topography with regional specificity. Furthermore, canonical correlation analysis reveals a shared pattern between the GS-behavior and respiration-behavior relationships, demonstrated as the linking between default mode network and psychiatric problems. In contrast, only GS topography correlates with cognitive performance. The reliability of respiration-GS relationships is confirmed via 10-fold cross-validated canonical correlation analysis. Additionally, this relationship is not replicated for another physiological signal, i.e., cardiac activity. Our findings underscore the functional and cognitive relevance of respiration to GS, rather than mere physiological noise. We propose the importance of considering respiration's multifaceted roles in modulating GS dynamics that underpin brain-body integration supporting mental health and cognitive function.

Resting-state fMRI studies often encounter the challenge in interpreting global brain activity. The global brain activity, also described as global signal (GS) in fMRI, is measured by averaging the activities across the whole brain[1–3]. Recent investigations have revealed a close relationship between brain arousal and GS, suggesting that the GS plays a role in regulating arousal levels[4–6]. Furthermore, the GS coordinates body-wide physiology in relation to anticipated behavioral demand[5,7–10]. In addition, the GS exhibits heterogeneous distribution across brain regions, resulting in a GS topography that reflects the non-uniform mapping onto local brain activity. At rest, higher GS topography are observed in the primary sensory cortex and lower levels in the higher-order cortical regions like prefrontal cortex[10–12], which are associated with psychiatric disorders like schizophrenia[13], bipolar disorder[14] and major depression disorder[15,16] as well as with board behavioral variables, including life outcomes and psychological function[12].

The origin and function of GS remain a topic of ongoing debate. While some studies have suggested subcortical contributions to GS[10,17–19], this represents just one perspective on its complex origins. Evidence suggests that GS may arise from multiple sources, including large-scale brain waves[7,20] and neuronal activity across distributed networks that associate with cognitive function and clinical relevance[8,21,22]. Other research indicates significant contributions from non-neuronal activities such as respiration, heartbeat, and blood transit effects[23–25]. This complexity has led to ongoing discussions about whether GS should be removed from functional Magnetic Resonance Imaging (fMRI) analysis[23,24].

Respiration is one major potential source of global signal, but its contribution to brain functions and behaviors remains unclear. On the physiological level, respiration changes lead to fluctuations in end-tidal $CO_2$ at a frequency of about 0.03 Hz, which significantly correlate with BOLD fMRI signal fluctuations[26–28]. Growing evidence confirms that these contributions of respiration to BOLD signal vary across different brain regions including both cortical and subcortical regions[9,11,26,29–31]. Independent of its effects on BOLD signals, respiration has been implicated in modulating various brain functions, including sensory processing, emotional regulation, and cognitive function[32–36]. It also influences state fluctuations such as arousal[7], trial-by-trial performance[37], and task-state changes[11]. Therefore, the idea that respiration is merely a nuisance factor for the BOLD signal may

[1]State Key Laboratory of Cognitive Neuroscience and Learning, Beijing Normal University, Beijing, China. [2]Institute for Neuropsychological Rehabilitation, University of Health and Rehabilitation Sciences, Qingdao, China. [3]School of Psychology, Chengdu Medical College, Chengdu, China. [4]Center for Brain Disorders and Cognitive Sciences, School of Psychology, Shenzhen University, Shenzhen, China. ✉e-mail: luoyj@bnu.edu.cn; zhangjf111@gmail.com

be overly simplistic. Understanding how respiration associates with GS and how both relate to cognition and behavior may provide insights into the physiological and functional aspects of the GS.

In this study, we investigated the functional relationship between respiration and GS and its behavioral relevance by comparing both spatial similarity and behavioral correlates in their topographies. We hypothesized that respiration had informative relationships with GS and its behavioral relevance, rather than being a nuisance factor. To test this hypothesis, we analyzed resting-state fMRI data from the Human Connectome Project ($N = 770$) using several complementary approaches. First, we computed topographic consistency between GS and respiration topographies using intraclass correlation (ICC)[38]. Results showed strong consistency in limbic and default mode networks, indicating regional specificity in the relationship between respiration and the GS. To examine behavioral relevance, we used canonical correlation analysis (CCA), a multivariate method of finding maximum correlation between linear combinations of two sets of variables[39,40]. This analysis revealed a shared pattern between GS-behavior and respiration-behavior relationships, demonstrated as the linking between default mode network and psychiatric problems. Additionally, we demonstrated that only the respiration-GS relationship, but not the heart-GS relationship, could reliably predict individual differences in behavior. These findings suggest that respiration's contribution to GS may have functional significance beyond mere physiological noise, particularly in relation to brain-body integration and behavioral outcomes.

## Results

In this study, we investigated whether respiration functionally related to global signal by comparing the spatial similarity and behavioral associations between global signal topography and physiological topography.

### Spatial patterns of GSCORR and RVTCORR and their similarities across networks

We first characterized the topography of global signal correlations (GSCORR) by correlating each region's time series with the mean global signal and applying Fisher's Z-transformation (Fig. 1A). The resulting GSCORR map replicated previous findings[10,11], demonstrating both robust strength (ranging from 0.0271 to 0.6484) and remarkable stability (topographic consistency between 2 days' rest: ICC = 0.9979, 95% CI = [0.9977, 0.9982]).

To elucidate the spatial contribution of physiological signals to global signal, we analyzed respiration volume per time (RVT), which captures both breathing rate and depth variations and exhibits established relationships with global signal fluctuations[30]. RVT demonstrated a positive shift relative to global signal, with the strongest negative correlation occurring at a time lag of 11.5 s using group-averaged cross-correlations[29] (Fig. 1B). This temporal relationship aligns with previous observations that fMRI signal increases follow decreases in respiration depth[26]. We then computed the topography of RVT correlations (RVTCORR) by correlating the time-shifted, flipped RVT signal with regional BOLD time series, to render RVTCORR comparable to GSCORR in subsequent analyses (Fig. 1B). While RVTCORR exhibited weaker absolute values (ranging from 0.0042 to 0.1132), it maintained high spatial stability (similarity between 2 days' rest: ICC = 0.9793, 95% CI = [0.9766, 0.9817]) (Fig. 2A).

The spatial relationship between RVT and GS, parameterized by intraclass correlation coefficient (ICC), showed moderate but consistent overlap between GSCORR and RVTCORR across participants ($M = 0.4481$, SD = 0.0868). This spatial correspondence demonstrated significant network-specific variation ($p < 0.0001$, Kruskal test[41]) (Fig. 1D). Notably, limbic, default mode, and salience networks exhibited stronger correlations, while visual and dorsal attention networks showed weaker associations.

### Overlap in the correlations of the GSCORR-behavior and RVTCORR-behavior relationships

Having established spatial relationships between GS and respiratory signals, we next investigated their behavioral correlates through canonical correlation analysis (CCA). CCA is a multivariate statistical method that simultaneously maximizes the correlation between two sets of variables. In this study, it was used to identify significant modes of co-variation between brain topography and behavioral measures across participants, after appropriate linear transformations[42]. Statistical significance was assessed through 5000 permutations that accounted for the family structure of the Human Connectome Project (HCP) data[12,42]. We identified one significant CCA mode (r = 0.7142, $p < 0.001$, Supplementary Fig. 1A) relating GSCORR to behavioral measures and two significant CCA modes (r = 0.7173, $p < 0.001$; r = 0.6852, $p = 0.0282$, Supplementary Fig. 1B, C) relating RVTCORR to behavioral measures.

**GSCORR-behavior relationship.** To examine the relationship between GSCORR and behavior in the principal CCA mode, we first correlated the individual GSCORR scores obtained from CCA first canonical mode with the original GSCORR map, resulting in 320 significant ROIs in the GSCORR-behavioral relationship ($p < 0.001$ for 10,000 permutations after controlling family-wise error rate (FWER)). The first mode demonstrated distinct spatial patterns: positive GSCORR weights were observed in the somatomotor network (SMN), the default mode network (DMN) and the salience network (SN), while negative weights emerged in the frontoparietal control network (CN) (Fig. 2A–C). The proportion of significant ROIs varied systematically across networks, with SMN showing the highest representation in the CCA related to behavior, while lower proportions were observed in other networks (Fig. 2C).

We then correlated the individual behavioral scores obtained from CCA first canonical mode with the original behavioral measures. This analysis identified 41 behavioral measures that demonstrated significant correlations with the principal mode ($p < 0.001$ for 10,000 permutations after FWER). The weights of these measures exhibited a clear positive-negative axis pattern, consistent with previous investigations of brain-behavior relationships[12,42]. Along this axis, positively correlated behavioral measures were associated with indicators of psychological distress, specifically psychiatric and emotional problems such as thought problems, antisocial personality problems, and childhood conduct problems. In contrast, negatively correlated measures predominantly reflected aspects of cognitive performance (Fig. 2E). To align with previous positive-negative axis[12,42], the weights of behavioral and topographic were inverted for visualization.

A strong correlation between behavioral and GSCORR scores was observed, as illustrated by thought problem scores—higher CCA scores of behavior or GSCORR corresponded to increased severity of thought problems, as demonstrated by the positive weight of thought problems within the GSCORR-behavior relationship (Fig. 2D).

**RVTCORR- behavior relationship.** We found that 178 ROIs and 22 behavioral measures showed significant correlations in the first RVTCORR-behavior mode. Within this mode, the RVTCORR weights were predominantly distributed in the default mode network (DMN) and the dorsal attention network (DAN) ($p < 0.001$ for 10,000 permutations after FWER) (Fig. 3A–C), a pattern that aligns with established research on respiration-modulated brain oscillations[35]. Analysis of behavioral correlates revealed two main categories of significant associations: first, physiological variables including substance abuse and bodily functions, and second, psychiatric problems such as thought problems, withdrawn problems, and childhood conduct problems. Notably, all psychiatric problems demonstrated significant negative correlations with individual behavioral scores from the RVTCORR-behavior CCA first canonical mode ($p < 0.001$ for 10,000 permutations after FWER) (Fig. 3E). For example, as behavioral scores and RVTCORR scores increased, the severity of thought problems decreased, reflected by the negative weight of thought problems within the RVTCORR-behavior relationship (Fig. 3D). The second mode of RVTCORR-behavioral relation identified 6 significant behavioral weights and 2 significant topographic weights, as detailed in Supplementary Fig. 2.

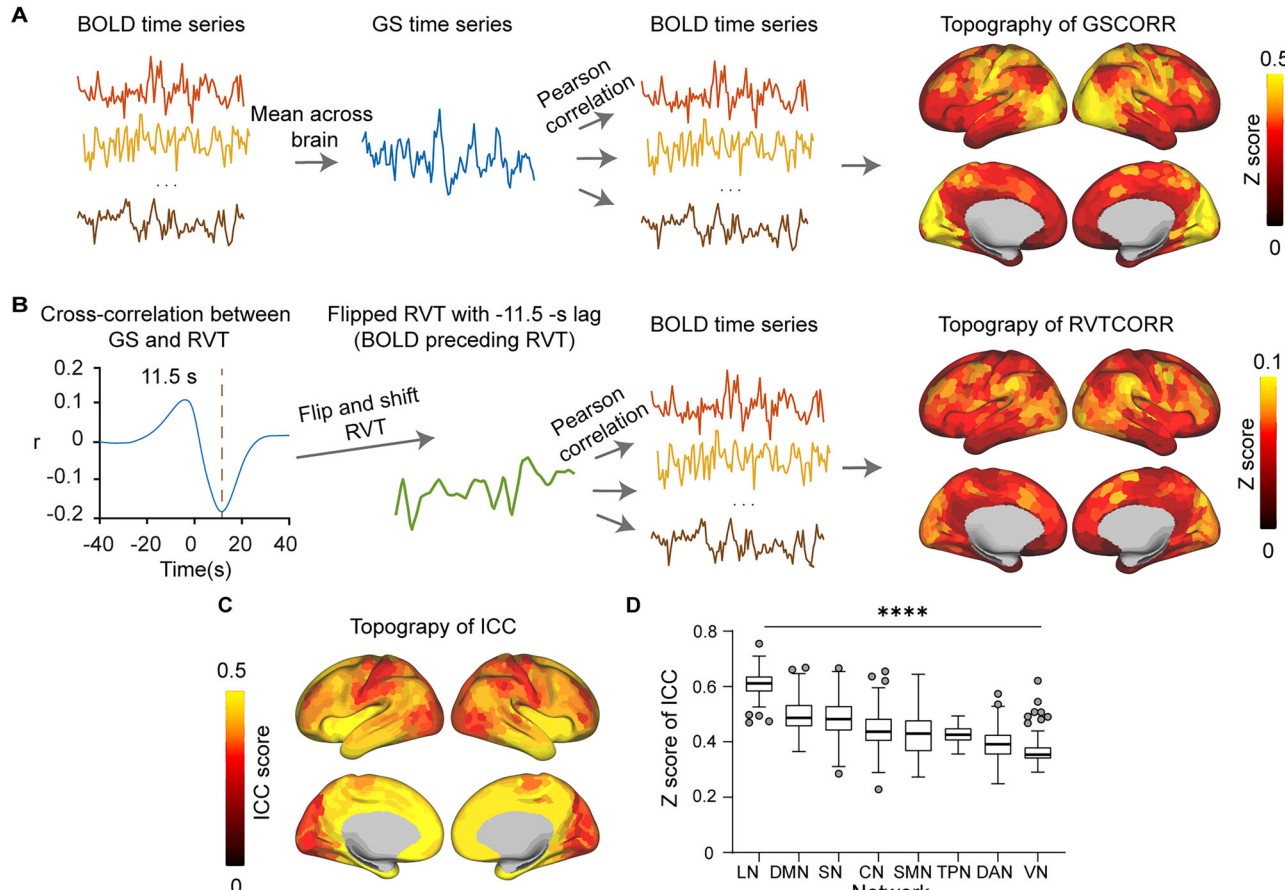

**Fig. 1 | Measurements of global signal and respiration topography. A** Steps to compute topography of GS correlations (GSCORR). GSCORR is computed as a Fisher's z-transformed Pearson's correlation coefficient between GS and time series in each region of interest (ROI) of the brain. **B** Steps to compute topography of respiration volume per time (RVT) correlations (RVTCORR). RVTCORR is computed as a Fisher's z-transformed Pearson's correlation between RVT and time series in each ROI of the brain. RVT is first flipped and shifted backward for 11.5 s lag to render RVTCORR comparable to GSCORR. **C** Topography of spatial consistency between GSCORR and RVTCORR. ROI-wise intraclass correlation coefficient (ICC) analysis is performed between GSCORR and RVTCORR across participants. **D** Box plot showing the Fisher's z-transformed ICC between GSCORR and RVTCORR grouped by networks ($n = 770$). Quantitative comparisons of ICC values among networks using Kruskal-Wallis test. **** $p < 0.0001$. LN limbic network, DMN default mode network, SN salience network, CN control network, SMN somatomotor network, TPN, temporal parietal network, DAN dorsal attention network, VN visual network. For source data, see Supplementary Data 1.

Given the established relationship between subcortical structures and respiration[9], we further tested subcortical contributions by recalculating CCA after regressing out subcortical activity from RVT. This analysis revealed diminished correlation strength and loss of significance in the RVTCORR-behavioral relationship ($r = 0.6797$, $p = 0.0618$, Supplementary Fig. 3), highlighting the importance of subcortical pathways in mediating respiratory effects on behavior.

**GSCORR-behavior versus RVTCORR-behavior.** As evident from the topographic and behavioral analyses, significant shared patterns emerged between GSCORR-behavior and RVTCORR-behavior relationships. The similarity between these two modes was quantified through correlations between their weights and individual scores[42]. Specifically, the first RVTCORR-behavior mode demonstrated robust correlations with the first GSCORR-behavior mode across multiple dimensions: behavioral scores ($r = 0.5372$), topographic scores ($r = 0.3660$), behavioral weights ($r = 0.5364$), and topographic weights ($r = 0.2831$). The opposite signs observed in the weights between the two modes likely resulted from eigenvector decomposition reversals during the CCA solution. To maintain consistency with previous findings, we reversed the sign of weights in the GSCORR-behavior pair[43]. We then visualized the overlapping significant ($p < 0.001$ for 10,000 permutations after FWER) topographic and behavioral weights with absolute values greater than 0.2 between first CCA mode of GSCORR-behavior and

RVTCORR-behavior pairs. This analysis revealed a shared pattern between both pairs. Specifically, both GSCORR and RVTCORR weights within the DMN (Fig. 4B) showed positive associations with psychiatric problems, including thought problems, antisocial personality problems and childhood conduct problems (Fig. 4A). In contrast, GSCORR uniquely exhibited weights in the CN that positively correlated with cognitive performance, a pattern which was absent in RVTCORR-behavior pairs.

To systematically examine these spatial contributions, we conducted regression analysis where RVTCORR was regressed out from GSCORR across the brain for each participant before performing the CCA. While a significant mode in the GSCORR-behavioral relationship persisted, we observed a decrease in the canonical correlation value ($r = 0.6853$, $p = 0.0284$) compared to the original analysis. More critically, this regression eliminated the previously observed overlapping significant ($p < 0.001$ for 10,000 permutations after FWER) behavioral weights with absolute values greater than 0.2 on the negative axis of the behavioral pattern, particularly those related to psychiatric problems such as thought problems, childhood conduct problems and antisocial personality problems (Fig. 4C). The significant ($p < 0.001$) overlap in the DMN was also reduced (Fig. 4D).

The second RVTCORR-behavior mode showed minimal correlations with the primary GSCORR-behavior mode across all dimensions: behavioral scores ($r = 0.0502$), topographic scores ($r = 0.0307$), behavioral weights ($r = 0.0382$), and topographic weights ($r = 0.0544$). Given these

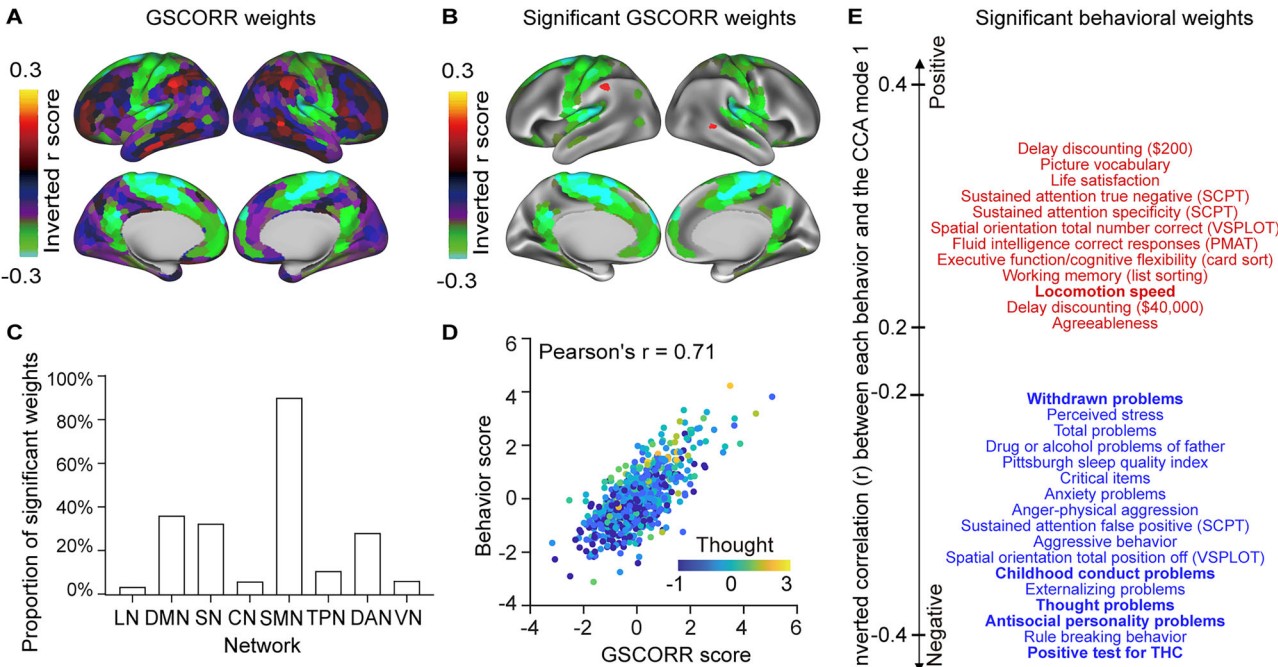

**Fig. 2 | Weights on the principal canonical correlation analysis (CCA) mode in GSCORR-behavior relationship. A** GSCORR weights from the first CCA mode. The signs of weights are inverted for visualization. **B** GSCORR weights that are significantly correlated with GSCORR scores ($n = 770$, $p < 0.001$, using 10,000 multiple permutations controlled for FWER, same for subsequent significance). The signs of weights were inverted for visualization. **C** Bar plot of significant GSCORR weights across networks, highlighting SMN as highest and LN as lowest. **D** Scatterplot of behavioral scores versus the GSCORR scores with an example behavioral measure (thought problems) with one point per participant in distinct color-coded data points. As behavioral score and GSCORR score increases, thought problems get more severe (lighter blue). **E** Significant ($p < 0.001$) behavioral weights

associated with the first CCA mode. Variables with absolute weight > 0.2 are shown, with red and blue indicating positive and negative weights, respectively. Bold font implies shared behavioral weights between GSCORR-behavior and RVTCORR-behavior pairs. The signs of weights were inverted for visualization. LN limbic network, DMN default mode network, SN salience network, CN control network, SMN somatomotor network, TPN temporal parietal network, DAN, dorsal attention network, VN visual network, SCPT Short Penn Continuous Performance Test, VSPLOT, Variable Short Penn Line Orientation Test, PMAT Penn Progressive Matrices Test, THC △⁹-tetrahydrocannabinol. For source data, see Supplementary Data 2.

weak associations, this secondary mode was not considered for further analysis.

### Reproducibility of CCA in GSCORR-behavior pair and RVTCORR-behavior pair

To evaluate the reproducibility of the identified canonical correlations, we implemented a rigorous 10-fold cross-validation scheme to assess model generalizability[40,44,45]. The data were systematically divided into 10 subsets with approximately equal numbers of participants, maintaining family structure integrity by keeping monozygotic twins from the same family within the same training set or test set. For each iteration, nine subsets served as the training set to derive pairwise topographic and behavioral weights, while the remaining subset functioned as the test set for computing out-of-sample correlation coefficients between individual topographic and behavioral scores based on the corresponding training set weights.

Table 1 displays the primary characteristics of the first mode for both GSCORR-behavior and RVTCORR-behavior pairs, encompassing generalizability of correlation coefficients and weight stability. The maximum out-of-sample canonical correlation coefficients reached 0.4170 and 0.4707 respectively, both achieving statistical significance after 5000 permutations ($p < 0.005$, Supplementary Fig. 4A, B). Moreover, the averaged out-of-sample correlations were 0.1017 and 0.2100, both demonstrating significance ($p < 0.05$, Supplementary Fig. 4C, D). These results provided robust evidence for reliable relationships between behavior and GSCORR, as well as between behavior and RVTCORR.

To demonstrate the robustness of CCA weights, we employed the same projection procedure on the split showing the highest out-of-sample correlation coefficient. Specifically, we correlated individual scores obtained from this best split with the original variables. Although the topographic and

behavioral weights of the GSCORR-behavior pair appeared on the opposite axis compared to previous results (Fig. 5A), this reversal was attributable to potential eigenvector decomposition reversals during CCA solution. The topographic and behavioral weights of the RVTCORR-behavior pair replicated our earlier findings (Fig. 5B). Notably, we observed consistent shared patterns between GSCORR-behavior and RVTCORR-behavior pairs, with both analyses demonstrating positive associations between GSCORR or RVTCORR weights in the DMN and psychiatric problems (Fig. 5C).

### The contribution of cardiac activity to GS and its behavioral relevance

After establishing the respiratory-GS relationships, we investigated whether cardiac activity demonstrated similar functional contributions to global signal. HR exhibited a distinct temporal relationship with global signal, showing a positive shift with maximum positive cross-correlation at a time lag of 0.72 s (Fig. 6A). This early peak in the cross-correlation aligned with previous research on cardiac-BOLD signal relationships[30,46].

Following the same analytical framework applied to respiratory signals, we computed HR topography (HRCORR) by correlating time-shifted cardiac signals with regional BOLD time series (Fig. 6B). HRCORR demonstrated weak but stable spatial patterns (range: −0.0017 to 0.0781), maintaining consistent topographic organization across sessions (spatial similarity between 2 days' rest: ICC = 0.9794, 95% CI = [0.9767, 0.9818]). The spatial contribution of HR to global signal was notably low (Mean = 0.0517, SD = 0.0425) and showed heterogeneous distribution across networks, with visual networks exhibiting the strongest similarity (Fig. 6C & D).

Most critically, CCA failed to identify any significant modes relating HRCORR to behavior measures (r = 0.6745, $p = 0.1956$, Fig. 6E). This

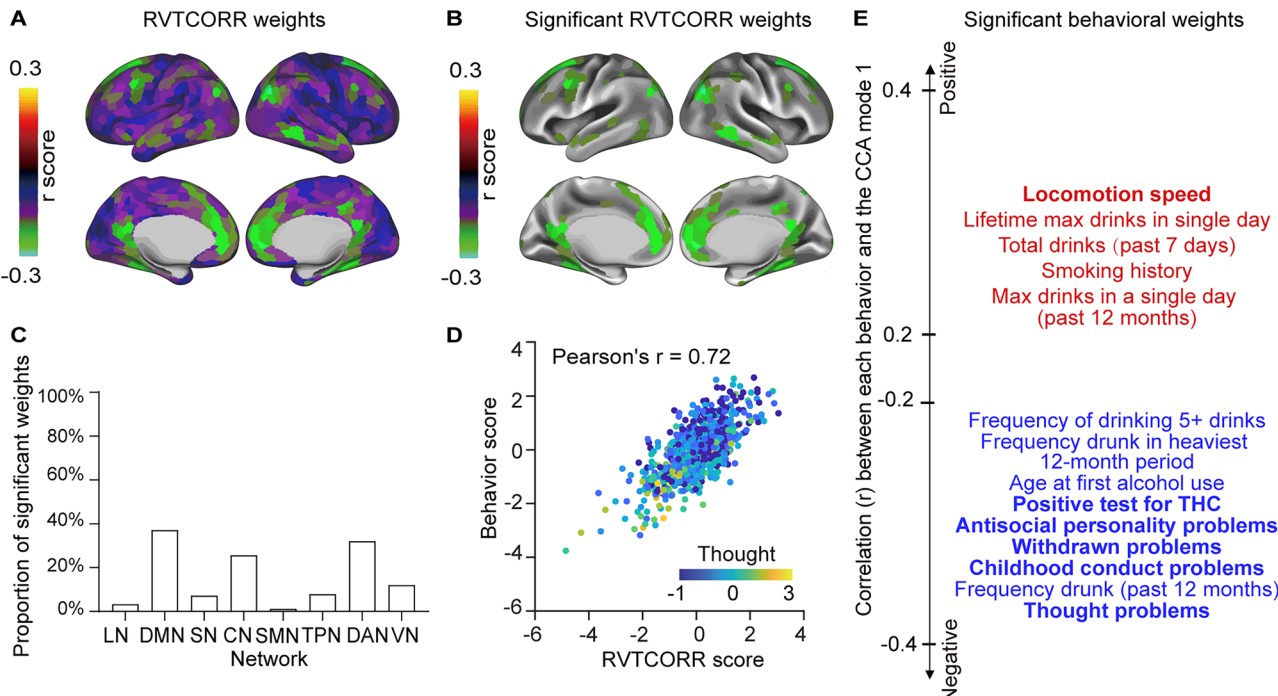

**Fig. 3 | Weights on the principal canonical correlation analysis (CCA) mode in RVTCORR-behavior relationship. A** RVTCORR weights from the first CCA mode. **B** RVTCORR weights that are significantly correlated with RVTCORR scores ($n = 770$, $p < 0.001$, using 10,000 multiple permutations controlled for FWER, same for subsequent significance). **C** Bar plot of significant RVTCORR weights across networks, showing highest in DMN and lowest in LN. **D** Scatterplot of behavioral scores versus RVTCORR scores with an example behavioral measure (thought problems) with one point per participant in distinct color-coded data points. As behavioral score and RVTCORR score increases, thought problems get less severe (darker blue). **E** Significant ($p < 0.001$) behavioral weights associated with the first CCA mode. Variables with absolute weight $> 0.2$ are shown, with red and blue indicating positive and negative weights, respectively. Bold font implying shared behavioral weights between GSCORR-behavior and RVTCORR-behavior pairs. LN limbic network, DMN default mode network, SN salience network, CN control network, SMN somatomotor network, TPN temporal parietal network, DAN dorsal attention network; VN visual network, THC $\triangle^9$-tetrahydrocannabinol. For source data, see Supplementary Data 3.

absence of significant behavioral correlations stood in marked contrast to the robust behavioral associations observed with respiratory signals, suggesting that the relationship between respiration and global signal extends beyond mere physiological confound to potentially serve functional roles (Fig. 6E).

These findings together demonstrate that while HR makes consistent contributions to global signal, these cardiac influences differ fundamentally from respiratory effects in both spatial organization and behavioral relevance. Unlike respiration, cardiac contributions to global signal cannot account for the observed global signal-behavior relationships, providing further evidence that the respiration-GS relationship possesses functional significance beyond methodological considerations.

**Control analyses**
We conducted a series of systematic validation analyses to rigorously evaluate the robustness of our findings across different methodological approaches.

The first validation examined the impact of preprocessing strategies by comparing minimal-preprocessing versus ICA-FIX (independent component analysis and a machine learning classifier) preprocessing results. In the minimal-preprocessing version, we observed stronger correlations between RVT and GS, evidenced by larger cross-correlation peak (Supplementary Fig. 5B). This version also demonstrated increased overall values for both GSCORR and RVTCORR (Supplementary Fig. 5A, C). We identified one principal significant CCA mode ($r = 0.7159$, $p < 0.001$, Supplementary Fig. 5E) that related GSCORR to behavioral measures and one principal significant CCA mode ($r = 0.6861$, $p < 0.05$, Supplementary Fig. 5F) that related RVTCORR to behavioral measures in the minimal-preprocessing version. Importantly, despite these preprocessing-dependent variations, the

core findings regarding topographic weights in DMN and their relationship with psychiatric problems remained consistent across both preprocessing approaches (Supplementary Fig. 5G, H) with high correlations of CCA patterns (see Tables 2–3). The HRCORR-behavior relationship remained non-significant in the minimal-preprocessing version ($r = 0.6721$, $p = 0.2542$, Supplementary Fig. 6).

Secondly, to evaluate contribution of subcortical structures to the CCA results, we incorporated an expanded template containing subcortical regions[47]. This analysis revealed significant modes in both GSCORR-behavior ($r = 0.7034$, $p < 0.001$, Supplementary Fig. 7D) and RVTCORR-behavior relationships ($r = 0.7032$, $p < 0.001$, Supplementary Fig. 7E) that closely aligned with our original findings, particularly in demonstrating the relationship between DMN topographic weights and psychiatric problems (Supplementary Fig. 7). The high correlations of individual CCA scores further supported the consistency of these results (see Tables 2–3). The HRCORR-behavior relationship was not significant ($r = 0.6766$, $p = 0.1356$).

Thirdly, to control the contribution of 0-timelag physiological signals to the brain activity, we regressed respiratory and cardiac signals with 0-time lag from the original time series, followed by GSCORR and RVTCORR computation and CCA from the residual time series. The first mode of GSCORR-behavior and RVTCORR-behavior relationships remained significant ($r = 0.7133$, $p < 0.001$; $r = 0.7150$, $p < 0.001$), with CCA results showing high similarity in both individual scores and weights to those in the original analysis (Tables 2–3).

Fourthly, to investigate the effect of the variance the RVT and GS shared over time on the CCA result, we employed two separate regression analyses. First, we regressed flipped RVT with a 11.5-s lag out of the GS, calculating GSCORR using residual GS and performed CCA. The

**A** Shared behavioral weights **B** Shared topographic weights

Locomotion speed
Positive test for THC
Childhood conduct problems
Antisocial personality problems
Thought problems
Withdrawn problems

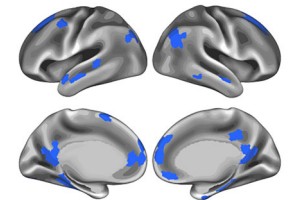

**C** Shared behavioral weights after RVTCORR regression **D** Shared topographic weights after RVTCORR regression

Locomotion speed
Positive test for THC

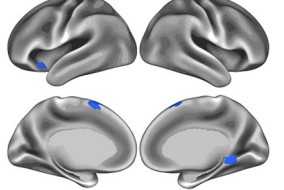

**Fig. 4 | Behavioral and topographic overlaps of the first canonical correlation analysis (CCA) modes in GSCORR-behavior and RVTCORR-behavior pairs.** **A** Overlap of significant ($n = 770$, $p < 0.001$, using 10,000 multiple permutations controlled for FWER, same for subsequent significance) behavioral weights with absolute values greater than 0.2. **B** Overlap of significant ($p < 0.001$) topographic weights. **C** Overlaps of significant ($p < 0.001$) behavioral weights with absolute values greater than 0.2 when GSCORR was regressed on RVTCORR prior to CCA. **D** Overlaps of significant ($p < 0.001$) topographic weights when GSCORR was regressed on RVTCORR prior to CCA. The signs of weights in the GSCORR-behavioral pair were inverted for visualization. The weights displayed here are significantly ($p < 0.001$) positively (red) or negatively (blue) associated with individual behavioral or topographic scores in sign-inverted GSCORR-behavior pair and RVTCORR-behavior pairs. For source data, see Supplementary Data 4.

GSCORR-behavior relationship remained significant ($r = 0.7141$, $p < 0.001$). Next, we regressed GS out of RVT, calculated the RVTCORR using residual RVT and performed CCA. The RVTCORR showed overall lower values (Supplementary Fig. 8B) and decreased spatial similarity with original GSCORR (Supplementary Fig. 8C). Although the RVTCORR-behavioral relationship persisted, we observed a decrease in the canonical correlation value ($r = 0.6873$, $p = 0.0122$) and reduction in overlapping behavioral weights related to DMN (Supplementary Fig. 8E) and psychiatric problems (Supplementary Fig. 8F). The result suggested that the RVT and GS did share topographic information over time, especially the relationship between DMN in RVTCORR and psychiatric problems.

Fifthly, to avoid the potential influence of group-level time lags overlooking individual differences, we calculated RVTCORR (Supplementary Fig. 9A) using individually optimized cross-correlation time lags (mean lag: $10.04 \pm 0.15$ s) and performed CCA. The spatial consistency between individual and group-level RVTCORR remained high (ICC: M = 0.9592, SD = 0.0918, Supplementary Fig. 9B). The RVTCORR-behavior relationship remained significant ($r = 0.7007$, $p < 0.001$) with high similarity to original patterns (Table 3).

Furthermore, to validate different measurement parameters of the respiratory signal, we calculated the alternative respiratory measures, including ENV and RV. CCA analyses of both the ENVCORR-behavior and RVCORR-behavior relationships produced comparable CCA scores and weights (Table 3), further supporting the robustness of our findings.

Lastly, to further explore the topographies in relation to cognitive tasks, we expanded the behavioral measures to additionally include 95 task-related behavioral variables and performed CCA. This yielded consistent patterns in both GSCORR-behavior and RVTCORR-behavior pairs (Tables 2–3), with preserved relationships between DMN in topographies and psychiatric problems (Supplementary Fig. 12).

In sum, this comprehensive validation framework demonstrated the stability of our findings across diverse methodological approaches and preprocessing strategies, strengthening the validity of our conclusions about the functional relationships between respiration, global signal, and behavior.

## Discussion

In this study, we investigated the functional relevance between respiration and global signal represented in the brain and its behavioral relevance. We observed that the GS and respiration showed a high spatial consistency, especially in limbic and default mode networks, suggesting that respiration and GS strongly shared brain representation with regional specificity. Moreover, the contribution was associated with behavioral significance. By using canonical correlation analysis, we observed similar behavioral associations between the GS topography and the RVT topography. Specifically, the default mode network association with psychiatric problems was observed in both the GS topography and the RVT topography. Further, canonical correlation analysis demonstrated that only the respiration-GS relationship, but not the HR-GS relationship, could reliably predict individual differences in behavior. This illustrates that the overlap representation of respiration and global signal in the brain is not noise, but has functional and cognitive significance. Moreover, our findings reveal novel insights into the potential links between brain representation of respiration and individual differences in behavioral phenotypes.

The global signal (GS) and its spatiotemporal dynamics are critical considerations in resting state fMRI analysis[2,20,23,24,48–50]. Inclusion or regression of the GS during preprocessing steps significantly impacts observed correlation patterns between networks[1,50–52]. The traditional perspective considers GS an artifact predominantly driven by non-neural factors like physiology and motion[23]. This view is supported by demonstrations of GS coupling to respiration, cardiac cycles, and apparent enhancement of network anti-correlations following GS regression[1,23,30,31]. As a result, the GS is often regressed out as noise.

However, culmulative electrophysiological evidence reveals significant GS correlations with neural activity, suggesting an informative neuronal basis[5,53,54]. Reliable topographical variations in GS also indicate functional relevance[5,8,11,14,16,20,55,56]. Our current results align with previous findings, confirming a close respiration-GS relationship with stable spatial consistency. Importantly though, Intraclass correlation coefficient analysis revealed heightened respiration-GS correlations specifically within limbic and default mode networks. This regional specificity implies respiration cannot fully explain the GS, with focused contributions to certain networks.

Of note, the respiration-GS relationship may dynamically vary across temporal lags, reflecting underlying physiological dynamics[30,31]. Future work should investigate potential temporal lag effects on this relationship. Overall, our findings provide clarification on the regional contributions of respiration to GS, going beyond a simplistic global correspondence model.

Respiration extensively shapes brain function and behavior. It synchronizes neural oscillations[33,57], alters arousal and perception[32,35,58,59], interacts with emotion and interoception[60–62], and provides oxygen enabling adaptive actions[63]. Respiration acts as a major source of fMRI variance through both vascular and neural effects[26,27,64]. Direct respiratory impacts on vascular tone and blood gases contribute to BOLD fluctuations and "physiological connectivity"[25,31,65–69]. Respiration also associates with brain activity changes through neural modulation of oscillations and vigilance-related regions[7,8,10,20,25,55,67–70].

During rest, GS variance tightly couples with respiration depth and rate[23,26,27]. However, this respiration-GS coupling noticeably attenuates during tasks[11], alongside observed cortical-subcortical anti-correlations[5,10,18]. This state-dependent dissociation indicates respiration does not fully account for GS dynamics. Instead, the GS represents an amalgam of global physiological and neural signals coordinating brain states like arousal[7,56]. Our observed respiration-behavior correlations further highlight potential functional roles for respiration in relation to GS, beyond mere noise. Overall, delineating the intricately multifaceted neural and physiological bases of GS

## Table 1 | Characteristics of the first CCA mode in GSCORR-behavior and RVTCORR-behavior pairs

| | Brain | Behaviors | Across-Modality Relationship | |
| --- | --- | --- | --- | --- |
| | Stability of weights [a] | Stability of weights [a] | In-Sample Correlation [b] | Out-of-Sample Correlation [c] |
| GSCORR | 0.7931 ± 0.0237 | 0.8313 ± 0.0206 | 0.7405 ± 0.0103 | 0.1017 ± 0.1852 |
| RVTCORR | 0.7997 ± 0.0435 | 0.8402 ± 0.0416 | 0.7360 ± 0.0069 | 0.2100 ± 0.1582 |

[a]Similarity of weights between each pair in training sets across 10 splits measured by Pearson's correlation.
[b]Averaged correlation between the canonical coefficients in the training sets across 10 splits.
[c]Averaged correlation between the canonical coefficients in the test sets across 10 splits.

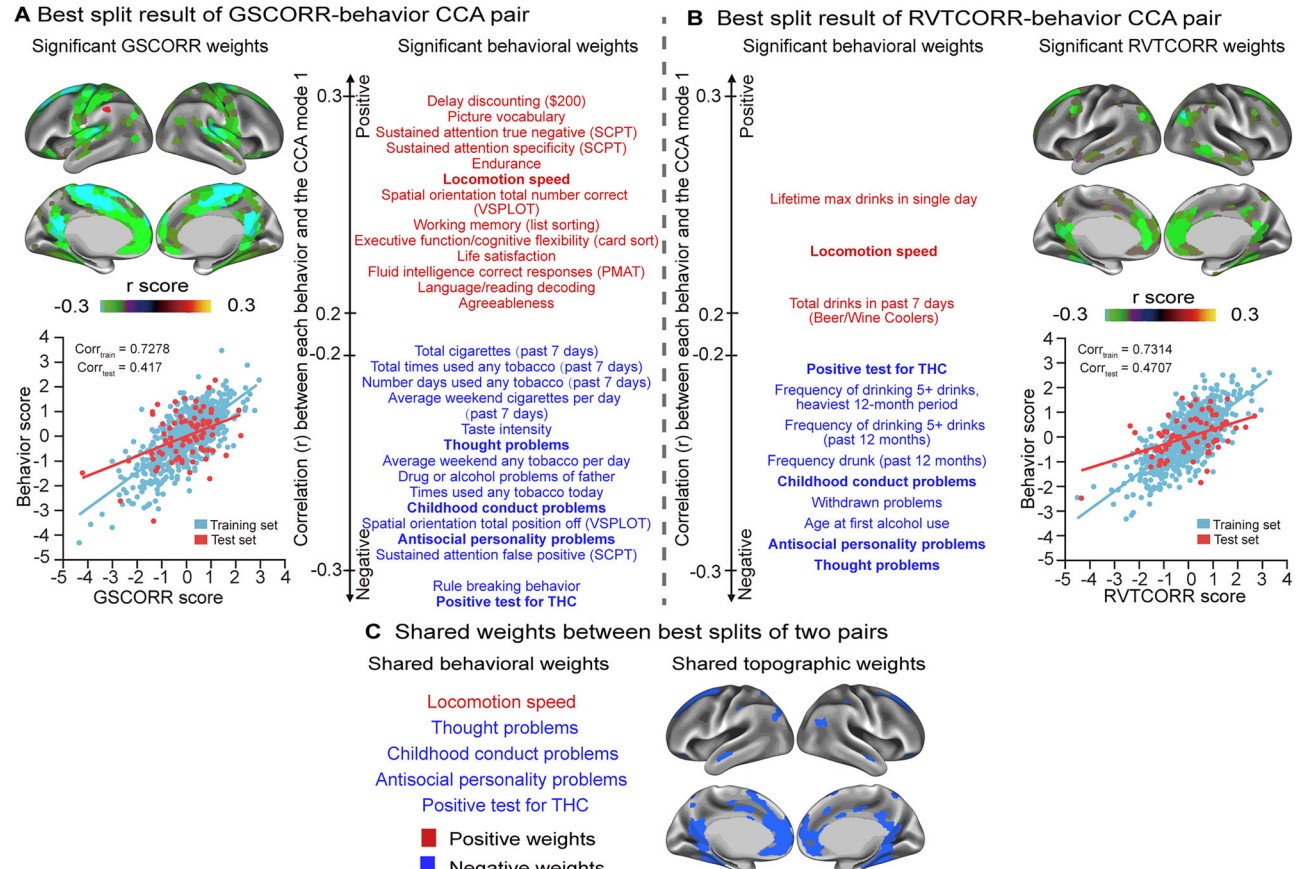

**Fig. 5 | Canonical correlation analysis (CCA) cross-validation of GSCORR-behavior pair and RVTCORR-behavior pair in the best-trained split. A** First CCA mode for GSCORR-behavior pair in the best-trained split. Upper panels: weights that are significantly correlated with scores of the first CCA mode ($n = 770$, $p < 0.001$, using 10,000 multiple permutations controlled for FWER, same for subsequent significance). Lower panel: scatterplot of behavioral scores versus GSCORR scores (train: blue; test: red). Each dot represents one participant. **B** First CCA mode for RVTCORR-behavior pair in the best-trained split. Upper panels: weights that are significantly correlated with scores of the first CCA mode ($n = 770$, $p < 0.001$). Lower panel: scatterplot of behavioral scores versus RVTCORR scores (blue: training; red: testing). Each dot represents one participant. **C** Overlaps of the first CCA mode between the GSCORR-behavior and RVTCORR-behavior pairs. Red and blue indicate positive and negative weights. SCPT Short Penn Continuous Performance Test, VSPLOT Variable Short Penn Line Orientation Test, PMAT Penn Progressive Matrices Test, THC $\Delta^9$-tetrahydrocannabinol. For source data, see Supplementary Data 5.

will require detailed assessment of respiratory dynamics across diverse brain states[8].

The dual-layer model (DLM) proposes that GS comprises interacting subcortical-cortical and cortical components, as well as physiological activity[8]. The subcortical-cortical background layer reflects global brain-body coupling supporting arousal regulation. The cortical surface layer features dynamic topography coordinating different forms of cognition.

Respiration directly links to the GS background layer. Aligned infra-slow fluctuations allow respiration to modulate vascular tone, metabolism, and arousal manifest in background GS dynamics[25,31,46,66–70]. However, the relationship between respiration and cortical GS topography appears more complex. During rest, respiration shapes topography, but this coupling decouples during tasks, indicating cortical topography is not a mere passive respiration projection.

Our results situate respiration's contributions to GS within the organizing DLM framework. Beyond arousal regulation, respiration may help drive infra-slow neural activity and interoceptive-exteroceptive processing underlying cognition and emotion. This could partially explain observed GS-behavior correlations. Further respiration-GS research can elucidate the multifaceted neural foundations and functions of respiration in brain dynamics.

CCA is a multivariate statistical method that identifies a relationship between two sets of variables[39]. One of its strengths is to identify common variation in two high-dimensional sets, targeting the prominent relationship

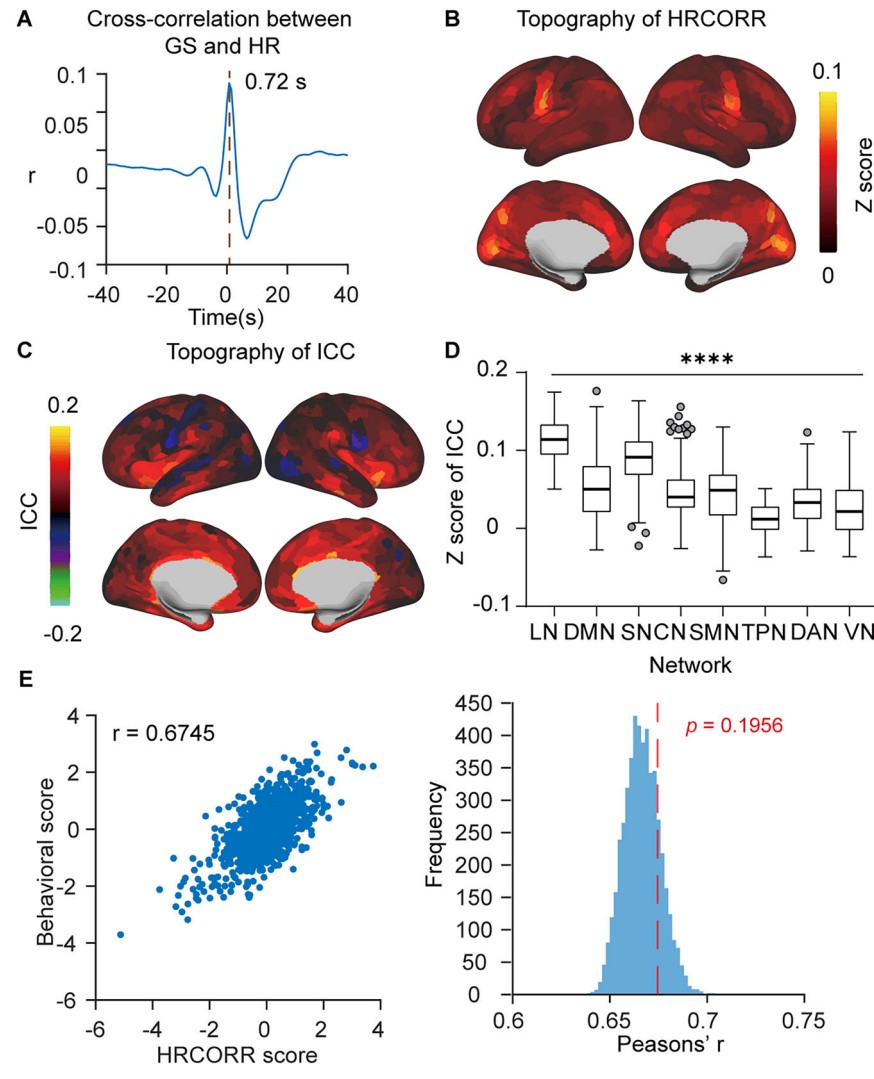

**Fig. 6 | Topography, spatial consistency of heart rate (HR) and canonical correlation analysis (CCA) with behavior measure. A** Cross-correlation between GS and HR. **B** Topography of heart rate correlations (HRCORR). **C** Spatial consistency between GSCORR and HRCORR, measured using intraclass correlation coefficient (ICC). **D** Box plot of Fisher's z-transformed ICC between GSCORR and HRCORR grouped by networks ($n = 770$). Quantitative comparisons of ICC values among networks are performed using Kruskal-Wallis test. **E** Scatterplot of behavioral scores versus HRCORR scores of first CCA mode, where each point represents one participant. **F** Null distribution of permuted canonical correlation coefficients (blue histogram) and the empirical canonical correlation coefficient (red dashed line) (right panel). ****$p < 0.0001$. LN limbic network, DMN default mode network, SN salience network, CN control network, SMN somatomotor network, TPN temporal parietal network, DAN dorsal attention network, VN visual network. For source data, see Supplementary Data 6.

shared across hundreds of variables. This makes it a particularly useful technique for links among brain, cognition, disease and genes[71]. Another unique advantage of multivariate approaches is to focus on inter-related patterns rather than unrelated single variables, which is suited for detecting complicated effects hidden in high-dimensional data sets[72]. Many researchers have studied the applications of CCA linking brain imaging modalities such as functional MRI, structural MRI and extended brain patterns including volume and density to behavioral and bodily measures that portray other aspects of individuals[12,42,73–75]. It is precisely because of the extensive impact of respiration on the brain[9] that we apply a multivariate approach to comprehensively explore the relationship between the effect of respiration on the brain and behavior. In our study, we observed that the similar brain representation contributed to behavior in CCA showed a critical difference from ICC result. The prominent consistent network in the ICC was located in the limbic network, while this seemed not significantly correlated with the behaviors in CCA. A possible explanation for this inconsistency could be the limitation of choice of time lag between respiration and BOLD signals, since respiration coordinates the limbic network dynamically underlying cognition processing[76]. The activity of limbic brain regions is likely to be dynamically influenced by respiratory control and perception[77]. Therefore, these respiration effects occur in a time-varying way rather than individual features.

The novelty of our work is the discovery of multimodal global respiration-brain phenotypes linked with psychiatric problems. Although previous work has shown that the DMN and SN were involved in

psychiatric problems[78–80], and higher respiration pattern variability and functional connectivity in DMN were related to depression[81], our approach allowed us to identify latent patterns characterized by the contribution of respiration to the brain in covariation with behavioral variables. Further work is needed to better understand multimodal, extended brain imaging and physiological recordings[73]. It remains unknown how diversely bodily rhythms dynamically influence brain patterns relating to individual difference in cognition and psychiatric problems[82].

We observed the connection between respiration, DMN, and psychiatric symptoms. This may likely involve multiple neurophysiological interacting pathways. Respiration influences brain activity through both direct and indirect mechanisms. Respiratory rhythms directly entrain neural oscillations across multiple brain regions[57,63], with particularly strong effects in limbic areas that interface with the DMN[34]. Additionally, respiration modulates arousal and interoceptive processing through bottom-up pathways from brainstem respiratory centers to cortical regions[33,61]. The DMN's strong involvement may reflect its role in integrating internal bodily states with self-referential processing and emotion regulation[83,84] - functions that are often disrupted in psychiatric conditions.

The link to psychiatric problems we observed could arise from disrupted respiratory-neural coupling affecting emotional and cognitive processing. The DMN is known to show altered activity patterns across multiple psychiatric conditions[85,86], and our results suggest that aberrant respiratory modulation of DMN activity could contribute to these disruptions. For example, anxiety and depression often involve disturbances in both

**Table 2 | Similarity of individual scores and weights between control analyses and primary CCA modes in GSCORR-behavior pair**

|   | Canonical correlation coefficient | Similarity of behavioral score | Similarity of topographic score | Similarity of behavioral weights | Similarity of topographic weights |
|---|---|---|---|---|---|
| 1 | 0.7159[***] | 0.7374 | 0.5904 | 0.7358 | 0.5575 |
| 2 | 0.7133[***] | 0.9974 | 0.9964 | 0.9974 | 0.9966 |
| 3 | 0.7034[***] | 0.7916 | 0.7192 |  |  |
| 4 | 0.7141[***] | 0.9845 | 0.9777 | 0.9843 | 0.9793 |
| 5 | 0.7265[***] | 0.7509 | 0.8455 |  |  |

[***]$p < 0.001$.
1: CCA performed with GSCORR constructed using the minimal-preprocessing version.
2: CCA performed with GSCORR constructed after regressing physiological signals out of original time series.
3: CCA performed with GSCORR constructed using the template that includes subcortical regions.
4: CCA performed with GSCORR constructed after regressing out RVT from GS.
5: CCA performed with extended behavioral variables measured during task-fMRI.

**Table 3 | Similarity of individual scores and weights between control analyses and primary CCA modes in RVTCORR-behavior pair**

|   | Canonical correlation coefficient | Similarity of behavioral score | Similarity of topographic score | Similarity of behavioral weights | Similarity of topographic weights |
|---|---|---|---|---|---|
| 1 | 0.6861[*] | 0.5570 | 0.4212 | 0.5505 | 0.3898 |
| 2 | 0.7150[***] | 0.9939 | 0.9915 | 0.9938 | 0.9915 |
| 3 | 0.7032[***] | 0.9050 | 0.8089 |  |  |
| 4 | 0.6961[**] | 0.5838 | 0.4708 | 0.5861 | 0.5126 |
| 5 | 0.7032[***] | 0.6421 | 0.5447 | 0.6450 | 0.5784 |
| 6 | 0.7007[***] | 0.6183 | 0.4768 | 0.6167 | 0.4678 |
| 7 | 0.7142[***] | 0.6637 | 0.7697 |  |  |

[*]$p < 0.05$, [**]$p < 0.01$, [***]$p < 0.001$.
1: CCA performed with RVTCORR constructed using the minimal-preprocessing version.
2: CCA performed with RVTCORR constructed after regressing physiological signals out of original time series.
3: CCA performed with RVTCORR constructed using the template that includes subcortical regions.
4: CCA performed with topography of envelope of the waveform.
5: CCA performed with topography of respiratory variation.
6: CCA performed with RVTCORR constructed by individually optimized cross-correlation time lags.
7: CCA performed with extended behavioral variables measured during task-fMRI.

breathing patterns and DMN function[9], potentially reflecting a cycle where altered respiratory-neural coupling affects emotional regulation and vice versa[81].

Moreover, recent work has demonstrated that respiratory dynamics can influence cognitive and emotional processing through multiple pathways[32,59]. These effects appear particularly pronounced in networks involved in emotional regulation and interoceptive awareness[35], suggesting a mechanistic link between respiratory patterns and psychological function.

However, we acknowledge that our correlational findings cannot establish causal relationships. Future research combining respiratory interventions with neuroimaging could help elucidate the precise mechanisms by which respiratory patterns influence DMN activity and subsequent behavior[87]. Understanding these pathways could have important implications for both psychiatric treatment and our broader understanding of brain-body integration in mental health[62].

We observed distinct behavioral relevance of respiration compared to cardiac activity, reflecting several key physiological and neurobiological mechanisms that warrant careful examination. Respiration exerts widespread effects on brain physiology through multiple pathways. Beyond direct effects on blood oxygenation and $CO_2$ levels influencing the BOLD signal[26,28], respiration entrains neural oscillations across distributed brain regions[63], particularly in limbic and default mode networks that showed strong overlap in our analyses. This entrainment occurs through both mechanical and chemical pathways[57,76]. In contrast, cardiac effects are primarily vascular[46] and more temporally confined, as evidenced by the early

time lag (0.72 s) we observed between cardiac and BOLD signals compared to respiratory effects (11.5 s lag).

Additionally, respiration is more intimately linked to arousal and cognitive states through interoceptive pathways and bottom-up modulation of brain activity[34,61]. The respiratory rhythm influences neural activity in regions involved in emotion[32], attention and behavioral control[35] processes that showed significant correlations in our behavioral analyses. While cardiac activity also has interoceptive effects[60], these appear more localized and may not have the same broad influence on behavioral networks that respiration demonstrates[33].

These mechanistic differences help explain why respiratory, but not cardiac, contributions to global signal showed reliable correlations with individual differences in behavior, particularly for psychiatric and emotional measures[81]. The differential temporal dynamics of respiratory (0.03 Hz) versus cardiac effects also suggest distinct mechanisms of influence on neural activity[29,31]. However, we acknowledge that future work using causal manipulation approaches[62] is needed to more fully characterize the neurobiological basis for these differential effects and their implications for understanding brain-body integration in health and disease[61,88].

Our results suggest that the relationship between respiration and global signal has functional significance rather than representing pure physiological noise[8,24], which has important implications for preprocessing approaches. Traditional preprocessing pipelines often treat respiratory signals as nuisance variables to be regressed out[1,23]. However, our findings that respiratory contributions to global signal correlate meaningfully with

behavior, particularly through default mode network activity and psychiatric measures[81,86], suggest this approach may remove functionally relevant neural signals. This is further supported by our observation that regressing out respiratory signals from global signal topography (GSCORR) significantly reduced behavioral correlations and eliminated previously observed relationships with psychiatric measures.

At the same time, we acknowledge that respiratory effects can include both neural and non-neural components[30,31]. Our comparison of FIX versus minimal preprocessing approaches[89] showed that while FIX removal of structured noise components reduced overall respiratory correlations, the key behavioral relationships remained intact. This suggests that careful noise removal can preserve functionally relevant respiratory-neural coupling[27] while reducing artifacts. Based on these findings, we suggest that rather than wholesale removal of respiratory signals, preprocessing pipelines might benefit from more nuanced approaches that consider the temporal delay between respiratory and BOLD signals[65], preserve global respiratory-neural coupling patterns while removing localized artifacts[90], and carefully evaluate the impact of preprocessing choices on behavioral correlations of interest. However, we acknowledge that optimal preprocessing strategies may vary depending on specific research questions and study designs[91]. Future work systematically comparing different approaches to respiratory signal handling could help establish more definitive best practices.

## Conclusion

This study provides novel evidence that respiration makes meaningful functional contributions to global signal that are relate to cognition and behavior. By comparing spatial and behavioral correlations of GS topography and respiration topography, we observed a shared pattern that links the default mode network to psychiatric problems. This challenges the view of respiration as mere physiological noise and instead highlights its informative relationship with the global brain, supporting mental health. Overall, delineating respiration's multifaceted neural bases will elucidate the complex amalgam of global physiological and neural signals coordinating brain-body integration and adaptive functioning.

## Methods

### Participants

The sample consisted of 770 participants (354 males; age range: 22-37 years) from the Human Connectome Project (HCP) S1200 release[92]. Resting-state functional MRI data were obtained from the HCP dataset, along with the respiration signals, cardiac signals and behavioral data[89,93]. Informed consent was obtained from all participants[92]. We excluded data from original 1100 participants based on the exclusion criteria indicated as follows: (1) missing entire rs-fMRI time series for any run ($n = 82$); (2) insufficient rs-fMRI time series for any run ($n = 15$); (3) loss of physiological recordings ($n = 87$); (4) insufficient numbers of triggers in physiological recordings ($n = 120$); (5) inability to perform reliable peak detection of the respiratory traces or reliable peak detection of the cardiac trace ($n = 24$); (6) participants without family structure ($n = 2$). The final sample comprised 770 participants.

### Resting-state fMRI and respiration recordings acquisition

Details of the HCP resting-state fMRI acquisition protocol was described elsewhere[92]. All participants were scanned on a 3-T Siemens connectome-Skyra scanner (customized to achieve 100 mT/m gradient strength) at Washington University in St. Louis[92]. Each participant underwent two sessions with two 15 min resting-state scans per session, utilizing a 32-channel head coil. Scanning parameters were: TR = 720 ms, voxel size = 2 mm isotropic. This resulted in four 1,200 sampled time points for each participant. During the resting state, participants were instructed to fixate on a crosshair, remaining awake with eyes open. Simultaneous cardiac and respiratory signals were recorded using pulse oximetry on a finger digit and a belt sensor around the abdomen, respectively, time locked to fMRI scan onset at a sampling rate of 400 Hz.

### Resting state-fMRI preprocessing

Data pre-processing step was implemented using Workbench 1.5.0[94] and custom codes in MATLAB 2020b[11]. The minimal preprocessing pipeline for the rs-fMRI has been described[95], including procedures of alignment using filed maps, advanced distortion correction methods, and anatomically informed registration algorithms, cleanup of structure temporal noise through independent component analysis and a machine learning classifier, nonlinear registration of T1w images to the MNI standard volumetric space, surface-based registration using multimodal surface matching Algorithm[89,95]. The HCP has adopted cleaning approaches including independent component analysis (ICA) and a machine learning classifier (FIX) to remove spatially specific structured artefacts and provided data of FIX-denoised version[30,89].

To elucidate the functional significance of respiratory effects beyond their impact on blood oxygenation[96], we primarily analyzed FIX-denoised data with supplementary analyses using the minimally-preprocessed data. The critical distinction between these approaches lies in ICA-FIX's capacity to remove spatially specific structured noise components, including: (1) spatial overlap with white matter, cerebrospinal fluid, or blood vessels; (2) signal localized at the edges of the brain (motion) or in areas of signal drop (susceptibility); (3) spatially ill-defined component clusters; (4) non-dominant low-frequency (< 0.1 Hz) spectral power; and (5) transient signal spikes. Crucially, recent studies has demonstrated that ICA-FIX method preserved spatially widespread "global" fluctuations[97] and GS[98] from fMRI signals which have shown strong relations to slow-frequency fluctuations of respiration and heart rate[30,99]. Overall, analyzing both versions ensured the genuine respiration contribution to the brain functions and behaviors.

Additional noise regression procedures were implemented through custom MATLAB code[11]. These included removal of linear trends for each run and regression of nuisance time series encompassing cerebrospinal fluid flow signals from ventricles, white matter signals, and head motion parameters. The regressors were derived from minimal preprocessing data. To control the contribution of 0-timelag physiological activity to brain activity, we evaluated versions incorporating the regression of respiration and cardiac signals, with results detailed in the Control Analyses section. For physiological signal integration, the original 400 Hz physiological recordings were temporally averaged to generate 1200-point time series matching the fMRI temporal resolution.

To enhance spatial signal-to-noise characteristics in local brain regions, analyses were conducted at the region-of-interest (ROI) level using a standardized brain parcellation comprising 998 ROIs[100]. This parcellation scheme generated fMRI time series with 1,200 sampled time points across four runs. Given established relationships between subcortical structures and global signal[10], we performed additional validation analyses incorporating a template comprising 416 ROIs, including 400 cortical regions and 16 subcortical regions[47], with results detailed in Supplementary Fig. 7.

### Preprocessing of physiological recordings

Respiratory and cardiac recordings were analyzed with the preprocessing followed the previous studies[11,23,26,27,30]. The respiratory signal was z-scored and linearly detrended. Outliers exceeding 3 median absolute deviations from the local median in a 0.3 s window were replaced via linear interpolation[30], i.e., 120 sampling time points using *filloutliers* function in MATLAB. It was then low-pass filtered at 5 Hz with a second-order Butterworth filter and z-scored[30]. Several measures haven been derived from the respiratory belt trace for the main results and control analyses, respectively, including the respiration volume per unit time (RVT), the windowed envelope of the waveform (ENV) and the respiration variation (RV)[101]. RVT was calculated as the difference between upper and lower signal envelopes over time[11]. ENV was calculated as the envelope of the trace over a 10 s window[101]. RV was calculated as the standard deviation of the trace within a 6 s window[101]. Waveforms were then resampled to 1200 time points to match fMRI sampling rate. The main results, followed the approach provided in our previous study[11], were based on RVT, as RVT was not influenced by the time window size[65].

Cardiac signals were initially z-scored and band-pass filtered with a second-order Butterworth filter between 0.3 and 10 Hz[30]. The peaks detected with a minimum distance of 0.3 s. Instantaneous heart rate (in beats per minute) was computed as 60 divided by inter-beat intervals. Outliers exceeding 3 median absolute deviations from the local 30 s median were replaced via linear interpolation.

## Calculation of GSCORR

The global signal was calculated by averaging time series across all regions of interest using the Schaefer parcellation, which specifically includes cortical gray matter regions[11]. For each participant at each scan, the GS topography was computed as the Pearson's correlation between the fMRI and GS time series, which were truncated by removing the first and last 10 time points to create a vector of 998 values[11]. These correlation values were then transformed into z-scores using Fisher's transformation. The resulting z-score vectors were computed separately for each 15 min run, and then averaged across four runs per participant, yielding a single GS topography (GSCORR) per participant.

## Calculation of RVTCORR and HRCORR based on cross-correlation with GS

RVT and HR time series were initially truncated by removing the first and last 10 time points. Considering the time delay between physiological signal and BOLD signal, we performed cross-correlation between GS and RVT, as well as GS and HR over the time lag from −72 s to 72 s, respectively. We examined the time lag corresponding to strongest absolute value of group-averaged cross-correlation coefficient to render physiological topographies comparable to GS topography in the subsequent analysis[11]. Regarding the individual variation in physiological responses, we also calculate the RVTCORR using individually optimized cross-correlation time lags, as shown in Supplementary Fig. 9. Considering that RVT ignored the effect of deep breathing[101], we also calculated the topography of ENV (ENVCORR) and topography of RV (RVCORR) according to their group-averaged time lag, respectively, and then performed the same follow-up analysis using ENV and RV, as shown in Supplementary Figs. 10, 11.

As a result, RVT time series were flipped and shifted backward by 16 time points (i.e., 11.5 s). HR time series were shifted backward by 1 time points (i.e., 0.72 s). We applied the same correlation steps as in GS topography to generate an averaged RVT topography (RVTCORR) and an averaged HR topography (HRCORR) per participant across scans.

## Intraclass correlation coefficient (ICC) analysis of GSCORR in comparison with RVTCORR and HRCORR

To quantify intra-individual variability between GSCORR and physiological topographies, we employed two-way random effect model[102], and calculated intraclass correlation coefficient (ICC) for each ROI between two topographies across participants in each scan. The ICC of λ is computed as:

$$\text{ICC}(\lambda) = \frac{\sigma_r^2}{\sigma_r^2 + \sigma_e^2}$$

where the variance component estimate $\sigma_r^2$ was derived by the difference between mean square rows and mean square error, and $\sigma_e^2$ was the mean square error. Then, we compared the proportion of ROI-wise ICC after averaging ICC across scans within networks[100] using rank-based test[41,103].

## Canonical correlation analysis (CCA)

In order to identify and compare the shared behavioral variables from GS and physiological topographies, we carried out canonical correlation analysis.

Before CCA, we did behavioral measures exclusion and data reduction for the original set of 478 behavioral measures as previously described[42]. We excluded 359 and kept 119 variables as following criteria: 1) 105 bad variables that were quantitatively poor measures including having 100 standard deviations above the median, fewer than half valid values (i.e., 500) or same

values exceeding 95% of the data. 2) 11 confounding variables including acquisition reconstruction software version, gender, age, weight, height, BMI, systolic blood pressure, diastolic blood pressure, Hemoglobin A1c measured in blood, the cube-root of total brain volume (including ventricles) and the cube-root of total intracranial volume. 3) uninterested variables in demographic measures. 4) redundant variables in cognition test and substance abuse retrospective. For more details, see Supplementary Note 1. The exclusion procedure described above resulted in 119 behavioral measures that were then fed into CCA. A list of employed behavioral measures is provided in Supplementary Data 7. For the primary results, we only used resting-state fMRI data and did not include task fMRI-based behavioral data in the initial CCA, which aligns with established protocols in the field for examining brain-behavior relationships and maintains methodological consistency. Furthermore, we extended our analysis to include additional 95 (initially 110, with 15 removed due to poor quality, including variables with values more than 100 standard deviations above the median, fewer than half of the values valid, or where more than 95% of the values were the same) behavioral variables measured in task-fMRI provided by HCP for additional information[104]. The names of additional task fMRI-based behavioral variables are provided in Supplementary Data 7.

To account for the missing behavioral measures (0.93% of data) and avoid overfitting in CCA, we estimated participants × participants covariance matrix and projected it onto the nearest valid (positive-definite) covariance matrix using the *nearestSPD* (http://www.mathworks.com/matlabcentral/fileexchange/42885-nearestspd) MATLAB toolbox[42] after normalizing and removing the confounds. Then we performed an eigenvalue decomposition of order 100 to obtain the top 100 eigenvectors of behavioral measures, which explained 99.53% of the total variance.

As for brain signals, the vector of 998-dimensional GSCORR, RVTCORR and HRCORR were conducted singular value decomposition after normalizing and removing the confounds, to derive 100-dimensional eigenvectors, explaining 80.57%, 89.16% and 89.86% of the total variance, respectively.

CCA as calculated by *canoncorr* function in MATLAB was then conducted on 100 behavioral and 100 topographic principal components, generating 100 modes of original canonical variate pairs ordered by canonical correlation coefficients and the corresponding 770 individual topographic scores and 770 individual behavior scores in each mode.

## Statistics and reproducibility

Statistical significance of each mode was estimated via 5000 permutations[105]. In each permutation, behavioral principal components and permuted topographic principal components by shuffling the order respecting the family structure in HCP were calculated CCA once to derive their first canonical variate pair. This resulted in a valid null distribution of 5000 maximal correlation values[42,106]. The empirical canonical variate pairs were declared statistically significant if their associated correlation coefficients exceeded the 95% percentile of null distribution (i.e., $p < 0.05$)[12]. In order to obtain relative weights of the original sets of topographic and behavior variables, we correlated the individual topographic and behavioral scores respectively against the 998-dimensional topographies and 119 behavioral variables across participants[42].

To map significant CCA modes onto original sets of topography, we correlated the topographic scores in the first mode with original 998-dimensional topography across participants via permutation correlation test with 10,000 permutations[12] based on Pearson's correlation, respectively, resulting in "full length" weights of topography. At the same time, we correlated the behavioral scores in the first mode with original 119 behavioral measures across participants via permutation correlation test with 10,000 permutations[12] based on Pearson's correlation, resulting in "full length" weights of behavioral measures. Statistical significance of the above correlation coefficient was set at $p < 0.001$ after controlling family-wise error rate (FWER)[42]. A weight with a significantly positive correlation coefficient contributed positively and reliably to the brain-behavior relationship obtained from the corresponding CCA mode, whereas a weight with a

significantly negative correlation coefficient contributed negatively and reliably to the brain-behavior relationship obtained from the corresponding CCA mode[74]. To identify potential overlap, we then compared the significant ($p < 0.001$) behavioral and topographic weights from the two different modes.

It is important to note that in CCA, as eigenvector decomposition may produce sign reversals that do not affect the magnitude of correlations between canonical variates[107]. The sign reversal of weights merely changes the direction of the correlation pattern without altering the magnitude of the correlation, as demonstrated in the proof provided in Supplementary Information. Throughout our analyses, we have adopted the conventional "positive-negative axis" established by Smith et al.[42] to facilitate interpretation consistency. According to this axis, behavioral canonical variables tend to positively correlate with favorable outcomes (cognitive abilities, education) while often negatively associating with less favorable outcomes (substance use, poor performance)[40,42,108,109]. In our analysis, we attempted to maintain consistency with the convention where variables such as "positive test for THC" might represent the "negative axis", which led us to adjust both behavioral and topographic weights accordingly. This approach does not alter the fundamental relationship patterns identified, but rather standardizes their directional visualization.

In order to evaluate the reproducibility of canonical correlation analysis, a 10-fold cross-validation scheme was performed to assess generalizability of the model[40,43–45].

First, we randomly split the data into 10 subsets, each containing approximately equal numbers of participants, ensuring that monozygotic twins from the same family remained in the same training set or test set[40]. For each split, we used nine subsets as a training set to perform PCA and CCA. For the remaining subset (the test set), we first applied the PCA weights derived from the training set to the test set data, followed by applying the CCA weights from the training set to the PCA-reduced test set data. Finally, we computed the Pearson's correlation between the resulting CCA vectors from the test set to obtain the out-of-sample correlation coefficient[40].

Next, to assess the statistical significance of the first CCA mode, we used 5000 permutations[43]. The $p$-values were calculated as the fraction of permuted out-of-sample correlation coefficients that exceeded the out-of-sample correlation coefficient obtained on the empirical data[40,44]. Given that the test set might be small, leading to potentially high variance in the correlation coefficients, the entire procedure above was repeated 10 times. To account for multiple comparisons, we applied a Bonferroni correction: the hypothesis will be rejected if any of the 10 split has a $p$-value less than 0.005[43,44]. To assess the generalizability of CCA model[44,45], we calculated the statistical significance of the averaged out-of-sample correlation value. Specifically, the $p$-values were determined as the fraction of permuted averaged out-of-sample correlation coefficients that exceeded the averaged out-of-sample correlation coefficient obtained on the empirical data.

To evaluate the reproducibility of CCA weights, we identified the split with the highest out-of-sample correlation coefficient. The stability of model weights was evaluated by averaging the pairwise Pearson's correlation of canonical weights across 10 splits[44,110]. Next, we projected the individual scores from the split back onto the original data and confirmed the behavioral and topographic overlaps.

### Similarity of patterns between the controlled and the original version

Using Pearson's correlation, we systematically evaluated the consistency of CCA patterns across different preprocessing strategies and analytical methods[42]. In general, each test is assessed based on four correlations, which are as follows: A) CCA Mode 1 individual behavioral scores from the controlled version versus the original scores; B) CCA Mode 1 individual topographic scores from the controlled version versus the original scores; C) CCA Mode 1 behavioral weights from the controlled version versus the original weights; D) CCA Mode 1 topographic weights from the controlled version versus the original weights.

### Reporting summary

Further information on research design is available in the Nature Portfolio Reporting Summary linked to this article.

### Data availability

The MRI and behavioral datasets including open and restricted data used in this study are available in the Human Connectome Project (S1200 release) repository, https://www.humanconnectome.org/study/hcp-young-adult/document/1200-subjects-data-release. All source data for graphs and charts in the main figures are available as Supplementary Data 1–6.

### Code availability

The scripts generated and used in the current study are available from the first or corresponding author on reasonable request.

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

## Acknowledgements

This work was supported by the Major Project of National Social Science Foundation (20&ZD153), National Natural Science Foundation of China (31920103009, 32201129); Shenzhen Hong Kong Institute of Brain Science – Shenzhen Fundamental Research Institutions (2023SHIBS0003); Shenzhen Science and Technology Program (20220811094132001), and the Start-up Research Fund in Shenzhen University.

## Author contributions

J.Y.: Conceptualization, Methodology, Formal analysis, Writing-original draft, Visualization, Writing - Review & Editing; J.F.: Conceptualization, Methodology, Writing - Review & Editing, Funding acquisition; Y.L.: Conceptualization, Supervision, Funding acquisition. All authors have seen and approved the final version of the manuscript being submitted.

## Competing interests

The authors declare no competing interests.

## Ethical approval

Informed consent was obtained from all participants[92]. The study has been conducted in accordance with Declaration of Helsinki. All ethical regulations relevant to human research participants were followed.
