## [Transparent Peer Review file · Communications Biology]

The Functional Overlap between Respiration and Global Signal and its Behavioral Relevance

Corresponding Author: Professor Jianfeng Zhang

Version 0:

Reviewer comments:

Reviewer #1

(Remarks to the Author)

The study by Yuan et al. investigated the functional relationship between respiration and the global signal (GS) in resting-state fMRI data. The authors found that respiration and GS exhibit strong spatial consistency and share a common pattern in their relationship with behavior, particularly in the default mode network and its association with psychiatric issues. Through canonical correlation analysis, they establish that both GS and respiration share behavioral relevance, emphasizing the necessity to consider physiological signals in the interpretation of fMRI data. These findings challenge the view that respiration is merely a nuisance factor in fMRI data and suggest that it has functional and cognitive significance. Overall, this is a well-written and informative paper that makes a significant contribution to our understanding of the relationship between respiration, global brain function, and behavior. The analysis is well-performed and the results are compelling. However, there are a few points that could be addressed to strengthen the paper further.

Major Comments:

The finding that respiration and GS share functional relevance raises an important question: To what extent does regressing out respiratory signals affect the measurement of brain-behavior relationships? Addressing this question could further elucidate the functional significance of respiration's contribution to brain activity. The authors may want to consider comparing brain-behavior correlations before and after respiratory signal regression to quantify its impact.

Furthermore, since both the global signal and the respiratory signal are one-dimensional time series, I wonder if there is a way to quantify the variance they share over time. Additionally, would it be worthwhile to examine the RVT CORR topography (as a control analysis) after regressing the global signal out from the respiratory signal?

The rationale for using CCA, as opposed to more traditional correlational methods for examining brain-behavior relationships, requires more detailed explanation. Additionally, the authors should justify their focus on only the first CCA mode. Are there potential insights to be gained from examining additional modes?

The authors apply the same time lag for both respiration and cardiac signals across all participants when computing their topographies. Why was a group-level approach chosen over individualized time lag selection? Given the potential for individual variability in physiological responses, would a personalized approach potentially yield more precise topographies?

Minor comments:

The implications of these findings for current fMRI preprocessing and analysis pipelines could be more thoroughly discussed. How might these results inform best practices for handling respiratory signals in resting-state fMRI studies?

While the authors demonstrate that cardiac activity does not show the same behavioral relevance as respiration, it would be valuable to discuss potential reasons for this difference. Are there physiological or neurobiological factors that could explain the unique contribution of respiration to GS and behavior?

The authors could expand the discussion on the potential mechanisms underlying the observed relationship between respiration, GS, and behavior. For instance, how might respiration influence neural activity in the default mode network, and how could this relate to psychiatric problems?

The authors utilized resting-state fMRI data from the Human Connectome Project, which, to my knowledge, includes data from more than 1,000 participants. They mentioned applying three exclusion criteria, resulting in valid data from 770 participants. I wonder if more details can be provided regarding the initial total number of participants and the number excluded per criterion.

In Line 217-218, it is stated, "Behavioral measures exclusion. To avoid the potential effect on our analysis, we did ..." It is unclear what specific potential effect the authors are referring to.

Reviewer #2

(Remarks to the Author)

The paper, "The Functional Overlap between Respiration and Global Signal and its Behavioral Relevance" by Yuan, Luo, and Zhang uses resting-state fMRI, respiration signals, cardiac signals and behavioral data from 770 young adult participants from the HCP S1200 release to investigate the correlations of global signal and respiration volume per unit time (and instantaneous heart rate), relationships with extensive behavioral measures, and their inter-relationships. Overall, the work was interesting, and it is important to better understand the underpinnings of global signal in rs-fMRI, but this work fell short in several ways:

1. It is unclear how/if quality control of the respiratory traces was performed. Based on the methods section, no participants were excluded (begins with "The sample consisted of 770 participants" and ends with "The final sample comprised 770 participants."). Based on recent publications using the same dataset (e.g., <https://doi.org/10.1038/s41467-020-18974-9>), I would be very surprised if no subjects were excluded specifically for an inability to perform reliable peak detection of the respiratory traces. Greater care in assessing the respiratory traces should be taken given this is the major contribution to the literature (nearly identical analyses using just the global signal were performed in <https://doi.org/10.1038/s41598-019-50750-8>).
2. The motivation for focusing on RTV (as opposed to RV or ENV) is not clear. Given the relatively weak correlation between RTV and rs-fMRI signal (Figure 1B), it would be worthwhile to investigate these other respiratory measures. This is especially relevant due to previous work demonstrating that RTV misses "deep-breaths" (<https://doi.org/10.1016/j.neuroimage.2019.116234>). Meanwhile, "bursts" are well captured by RTV (potentially biasing the current paper's results), occur more often in males, and are believed to be linked to chemoreflex-driven clinical breathing patterns that have neurological/psychiatric/medical associations which could potentially explain the current paper's RVTCCORR CCA results.
3. A main aim of the paper is to investigate shared features between global signal and respiration as they relate to region-specific rs-fMRI signal and subsequently behavior. It would therefore be beneficial to more thoroughly test for the behavioral and topographic overlaps described in Figure 4. Currently, there does not appear to be any statistical tests associated with the overlaps or for the observation that the overlap is predominantly in the DMN.
4. The 10-fold CV CCA was not performed in a robust or convincing manner. Given the growing literature suggesting the need for larger sample sizes in these types of analyses (<https://doi.org/10.1038/s42003-024-05869-4>) and, at the very least, sufficient cross-validation (<https://doi.org/10.1038/s41586-023-05745-x>), I appreciate the authors attempting to perform cross-validation of the CCA. But taking the "best split result" is not a robust way of performing cross-validation as this will inflate the results—similar (but to a lesser degree) to the in-sample results—so typically the average of the CV results is used. In this case, I don't believe the out-of-sample performance of either GSCORR or RVTCCORR would be significant (although the null distribution from the permutation test would be different since you would be averaging across the 10-fold CV permuted correlations). Because the in-sample GSCORR CCA results themselves are not novel (<https://doi.org/10.1038/s41598-019-50750-8>), more robust testing of the out-of-sample performance is needed.
5. Related to the 10-fold CV CCA: the HCP dataset consists of many related individuals. It is important to account for this in the CV as the inclusion of related individuals (e.g., twins) in the training and test sets will inflate the out-of-sample performance.
6. I'm a bit confused as to why the authors used ICC to compare the GSCORR and RVTCCORR maps across subjects. This is typically used as a measure of consistency across raters or measurements. Why not measure Pearson's correlation instead?
7. Greater detail is needed throughout the methods section.

Reviewer #3

(Remarks to the Author)

Major Comments:

line 58-62: The author discusses subcortical contributions to the 'generation' of the global signal. While this is valid, it is not

the only possible explanation. Turchi and colleagues have proposed this idea, but I find the reasoning unclear, especially when considering the possibility that the global signal could also be noise and other sources of signal. Please rephrase this statement and provide additional evidence regarding the origin of the global signal observed in the resting state, which could also stem from other sources, such as brain waves (Pang et al., 2023; Roberts et al., 2019)

line 68-86: While the information provided in this section is useful to the reader, the overall reasoning seems confusing. Is the author trying to explain the physiological meaning of the global signal by suggesting that, since breathing is correlated with emotional states, and the global signal is correlated with breathing, the global signal reflects breathing rather than brain function? If so, please rephrase the sentence in way it is more comprehensible.

line 92-106: The author has listed all the results and conclusions in this section, but some strong statements are left unjustified. I strongly suggest focusing on detailing the study's aim and implications, and then summarizing the main results to confirm or refute the hypothesis.

line 146: In the introduction, the author suggests that the global signal (GS) may originate from subcortical structures, while the Schafer parcellation typically does not include subcortical parcellation. I strongly recommend including subcortical structures (Tian et al., 2020) in the GS computation. Furthermore, these subcortical structures are also closely related to classical regions associated with respiratory signals.

line 138: Using FIX to investigate the relationship between respiration and the GS seems problematic. Perhaps I missed the rationale, but FIX typically regresses out signals related to respiration and CSF (Griffanti et al., 2017). Please provide the rationale for this approach, along with an analysis both with and without FIX applied, which would involve preprocessing the HCP data in a different manner.

line 217: Excluding some behavioural data from the analysis for statistical reasons seems reasonable. However, it might be more meaningful to exclude or include tasks based on their potential to provide insights into the physiological significance of the GS. Provide the rationale for the behavioural data.

Minor Comments:

line 40-41: please, specify if the Global Signal (GS) is related to white matter – grey matter or both.

line 48-50: this sentence needs a citation

line 122 gradient strength of 300 mT/m (millitesla per meter)

line 198: the participants were 770?

Version 1:

Reviewer comments:

Reviewer #1

(Remarks to the Author)

I thank the authors for their careful revision. My previous concerns have been fully addressed.

Reviewer #2

(Remarks to the Author)

I'd like to thank the authors for their detailed responses to the reviewers' comments. I find the revised manuscript to be much improved. Below are my remaining comments:

1. It's not entirely clear to me why the authors chose to reverse the sign of the CCA weights for the GSCOR analyses. I understand that it is possible that CCA reverses the weights (as referenced by the authors here "To maintain consistency with previous findings, we reversed the sign of weights in the GSCORR-behavior pair64), but the motivation for choosing to reverse the weights is not substantiated in the current text. Please provide references and justification for choosing to reverse the sign of CCA weights for the GSCORR analyses (as opposed to reversing the sign of the CCA weights for the RVTORR analyses), make it clear what criteria was used to decide whether to flip the sign or not (e.g., behavioral weight for THC should always be negative), and regenerate all of the figures to maintain consistency of the weight directions (i.e. reverse the signs of the weights for Figure 2, Supplementary Figure 7, Supplementary Figure 11).

2. I found it very interesting that the shared topographic weights between the first CCA modes in the GSCORR-behavior and RVTORR-behavior pairs encompassed a much larger portion of the brain than when including the task-fMRI behavioral variables (Supplementary Figure 12) compared to when the task-fMRI behavioral variables were not included (Figure 4). Could the authors discuss why they think this might be the case and explain why they chose not to include these variables from the start?

3. Please provide all of the panels from Figure 2 and Figure 3 when including the tasked-based behavioral variables in the CCA for completeness.

4. Please make the naming of the plots of the behavioral weights consistent across all of the figures (e.g., Supplementary Figure 10 uses the raw variable names whereas Figure 2 uses a more conventional naming).

5. Lines 713–714: "Both relationship of ENVCORR-behaviors (Supplementary Figure 10) and relationship of EVCORR-behaviors (Supplementary Figure 11)" should be "Both relationship of ENVCORR-behaviors (Supplementary Figure 10) and

relationship of RVCORR-behaviors (Supplementary Figure 11).”

Reviewer #3

(Remarks to the Author)

I would like to thank the author for addressing all of my concerns; I have nothing further to add.

REVIEWERS' COMMENTS:

Reviewer #1 (Remarks to the Author):

The study by Yuan et al. investigated the functional relationship between respiration and the global signal (GS) in resting-state fMRI data. The authors found that respiration and GS exhibit strong spatial consistency and share a common pattern in their relationship with behavior, particularly in the default mode network and its association with psychiatric issues. Through canonical correlation analysis, they establish that both GS and respiration share behavioral relevance, emphasizing the necessity to consider physiological signals in the interpretation of fMRI data. These findings challenge the view that respiration is merely a nuisance factor in fMRI data and suggest that it has functional and cognitive significance. Overall, this is a well-written and informative paper that makes a significant contribution to our understanding of the relationship between respiration, global brain function, and behavior. The analysis is well-performed and the results are compelling. However, there are a few points that could be addressed to strengthen the paper further.

Major Comments:

The finding that respiration and GS share functional relevance raises an important question: To what extent does regressing out respiratory signals affect the measurement of brain-behavior relationships? Addressing this question could further elucidate the functional significance of respiration's contribution to brain activity. The authors may want to consider comparing brain-behavior correlations before and after respiratory signal regression to quantify its impact.

Reply: *We agree that examining the impact of respiratory signals on brain-behavior relationships through regression analysis is crucial for understanding their functional significance.*

When respiratory and cardiac signals were regressed with 0-time lag from the original time series (line 152-156), we found that the GSCORR-behavior relationship remained significant ($r = 0.7133$, $p < 0.001$) (line 682-688) and the CCA results maintained high similarity in both individual scores and weights to the original analysis, as shown in Table 2 (line 729-737). These findings suggest that direct regression of concurrent physiological signals does not substantially alter the primary brain-behavior relationships we observed, likely due to the temporal dynamics of physiological effects on brain activity.

Furthermore, to examine spatial contributions systematically, we conducted regression analysis where RVTCORR was regressed from GSCORR across the brain for each subject before performing CCA. While a significant mode in the GSCORR-behavioral relationship was maintained, we observed a notable decrease in the canonical correlation value ($r = 0.6853$, $p = 0.0284$) compared to the original analysis (line 520-530). More critically, this regression eliminated the previously observed overlapping behavioral weights on the negative axis of the

behavioral pattern, particularly those related to psychiatric problems such as thought problems and antisocial personality problems. The attenuation of both behavioral and topographic weights following RVT CORR regression, especially in the default mode network, provides quantitative evidence for RVT CORR's meaningful contribution to the GSCORR-behavior relationship. The regression results showing reduced overlap in both topographic and behavioral weights are presented in Figures 4C-D.

Furthermore, since both the global signal and the respiratory signal are one-dimensional time series, I wonder if there is a way to quantify the variance they share over time. Additionally, would it be worthwhile to examine the RVT CORR topography (as a control analysis) after regressing the global signal out from the respiratory signal?

Reply: *Thank you for these important methodological questions. Regarding temporal variance they shared, the information can be found in our cross-correlation (Figure 1B). Our cross-correlation analysis demonstrated a consistent relationship where respiratory effects preceded global signal changes by approximately 11.5 seconds, as shown in Figure 1B of the manuscript. This temporal precedence aligns with established physiological models of respiratory-BOLD coupling.*

To investigate the effect of the variance the RVT and GS shared over time on the CCA result, we employed two separate regression analyses. First, we regressed flipped RVT with a 11.5-s lag out of the GS, calculating GSCORR using residual GS and performed CCA. The GSCORR-behavior relationship remained significant ($r = 0.7141$, $p < 0.001$). Next, we regressed GS out of RVT, calculated the RVT CORR using residual RVT and performed CCA. The RVT CORR showed overall lower values (Supplementary Figure 8B) and decreased spatial similarity with original GSCORR (Supplementary Figure 8C). Although the RVT CORR-behavioral relationship persisted, we observed a decrease in the canonical correlation value ($r = 0.6873$, $p = 0.0122$) and reduction in overlapping behavioral weights related to DMN (Supplementary Figure 8E) and psychiatric problems (Supplementary Figure 8F) (line 690-702).

The rationale for using CCA, as opposed to more traditional correlational methods for examining brain-behavior relationships, requires more detailed explanation. Additionally, the authors should justify their focus on only the first CCA mode. Are there potential insights to be gained from examining additional modes?

Reply: *The rationale for employing CCA stems from its unique ability to identify complex multivariate relationships between high-dimensional datasets, making it particularly well-suited for examining brain-behavior associations that may not be captured by traditional univariate correlations. As detailed in our methodology section, CCA enabled us to simultaneously analyze relationships between 998-dimensional brain topography measures and 119*

behavioral variables, identifying optimal linear combinations that maximize correlations between these datasets.

Regarding the question about additional CCA modes, we did identify a second significant mode in the RVT CORR-behavioral relationship ($r = 0.6852$, $p = 0.0282$) (line 404-407; line 470-472). However, when comparing this mode to the first CCA mode of the GSCORR-behavioral relationship, we found low similarity across multiple parameters as quantified in Table 2 of the manuscript. Specifically, the second RVT CORR-behavior mode showed minimal correlations in behavioral scores ($r = 0.0502$), topographic scores ($r = 0.0307$), behavioral weights ($r = 0.0382$), and topographic weights ($r = 0.0544$) with the first GSCORR-behavior mode (line 532-536). Given our focus on understanding shared variance between respiratory and global signal contributions to behavior, and the weak correspondence between this second mode and our primary findings, we chose to concentrate our analysis on the first canonical mode. The complete results for the second RVT CORR mode are also provided in Supplementary Figure 2 of the supplementary materials.

The authors apply the same time lag for both respiration and cardiac signals across all participants when computing their topographies. Why was a group-level approach chosen over individualized time lag selection? Given the potential for individual variability in physiological responses, would a personalized approach potentially yield more precise topographies?

Reply: *Our choice of a group-level approach for physiological time lag determination was driven by methodological considerations regarding signal-to-noise optimization. While individual variations in physiological responses are important, the group-averaged approach provided more stable cross-correlation curves compared to individual subject analyses. Nevertheless, we systematically evaluated the potential impact of individualized time lags (line 206-208; line 704-710).*

Specifically, we computed RVT CORR using subject-specific cross-correlation time lags, which yielded a mean lag of 10.04 ± 0.15 seconds (group mean \pm standard error). To quantify the consistency between individualized and group approaches, we calculated the intraclass correlation coefficient (ICC) between RVT CORR maps derived from individual versus group delays. This analysis revealed high spatial consistency ($M = 0.9592$, $SD = 0.0918$), as documented in Supplementary Figure 9B of the supplementary materials.

These validation analyses suggest that while individual variability exists in physiological responses, our group-level approach captured the predominant temporal relationships without significantly compromising the detection of behaviorally relevant brain-physiology associations.

Minor comments:

The implications of these findings for current fMRI preprocessing and analysis pipelines could be more thoroughly discussed. How might these results inform best practices for handling respiratory signals in resting-state fMRI studies?

Reply: Thank you for raising this important methodological question about the implications of our findings for fMRI preprocessing. We have now included the section of discussion below: Our results suggest that the relationship between respiration and global signal has functional significance rather than representing pure physiological noise (Liu et al., 2017; Zhang & Northoff, 2022), which has important implications for preprocessing approaches.

Traditional preprocessing pipelines often treat respiratory signals as nuisance variables to be regressed out (Murphy & Fox, 2016; Power et al., 2017). However, our findings that respiratory contributions to global signal correlate meaningfully with behavior, particularly through default mode network activity and psychiatric measures (Zamoscik et al., 2018; Sheline et al., 2009), suggest this approach may remove functionally relevant neural signals. This is further supported by our observation that regressing out respiratory signals from global signal topography (GSCORR) significantly reduced behavioral correlations and eliminated previously observed relationships with psychiatric measures.

At the same time, we acknowledge that respiratory effects can include both neural and non-neural components (Chen et al., 2020; Kassinosopoulos & Mitsis, 2019). Our comparison of FIX versus minimal preprocessing approaches (Glasser et al., 2016) showed that while FIX removal of structured noise components reduced overall respiratory correlations, the key behavioral relationships remained intact. This suggests that careful noise removal can preserve functionally relevant respiratory-neural coupling (Birn et al., 2008) while reducing artifacts. Based on these findings, we suggest that rather than wholesale removal of respiratory signals, preprocessing pipelines might benefit from more nuanced approaches that consider the temporal delay between respiratory and BOLD signals (Chang et al., 2009), preserve global respiratory-neural coupling patterns while removing localized artifacts (Power et al., 2018), and carefully evaluate the impact of preprocessing choices on behavioral correlations of interest. However, we acknowledge that optimal preprocessing strategies may vary depending on specific research questions and study designs (Caballero-Gaudes & Reynolds, 2017). Future work systematically comparing different approaches to respiratory signal handling could help establish more definitive best practices.

While the authors demonstrate that cardiac activity does not show the same behavioral relevance as respiration, it would be valuable to discuss potential reasons for this difference. Are there physiological or neurobiological factors that could explain the unique contribution of respiration to GS and behavior?

Reply:

Thank you for this insightful question about the differential contributions of respiratory versus cardiac signals to behavior. We have now included the section of discussion below

The differential contributions of respiratory versus cardiac signals to behavior

We observed distinct behavioral relevance of respiration compared to cardiac activity, reflecting several key physiological and neurobiological mechanisms that warrant careful examination.

Respiration exerts widespread effects on brain physiology through multiple pathways. Beyond direct effects on blood oxygenation and CO₂ levels influencing the BOLD signal (Birn et al., 2008; Wise et al., 2004), respiration entrains neural oscillations across distributed brain regions (Tort et al., 2018), particularly in limbic and default mode networks that showed strong overlap in our analyses. This entrainment occurs through both mechanical and chemical pathways (Heck et al., 2017; Karalis & Sirota, 2022). In contrast, cardiac effects are primarily vascular (Chang et al., 2009) and more temporally confined, as evidenced by the early time lag (0.72s) we observed between cardiac and BOLD signals compared to respiratory effects (11.5s lag).

Additionally, respiration is more intimately linked to arousal and cognitive states through interoceptive pathways and bottom-up modulation of brain activity (Critchley & Harrison, 2013; Zelano et al., 2016). The respiratory rhythm influences neural activity in regions involved in emotion (Boyadzhieva & Kayhan, 2021), attention and behavioral control (Kluger et al., 2021) - processes that showed significant correlations in our behavioral analyses. While cardiac activity also has interoceptive effects (Park & Blanke, 2019), these appear more localized and may not have the same broad influence on behavioral networks that respiration demonstrates (Varga & Heck, 2017).

These mechanistic differences help explain why respiratory, but not cardiac, contributions to global signal showed reliable correlations with individual differences in behavior, particularly for psychiatric and emotional measures (Zamoscik et al., 2018). The differential temporal dynamics of respiratory (0.03 Hz) versus cardiac effects also suggest distinct mechanisms of influence on neural activity (Yuan et al., 2013; Chen et al., 2020). However, we acknowledge that future work using causal manipulation approaches (Nord & Garfinkel, 2022) is needed to more fully characterize the neurobiological basis for these differential effects and their implications for understanding brain-body integration in health and disease (Critchley & Garfinkel, 2018).

The authors could expand the discussion on the potential mechanisms underlying the observed relationship between respiration, GS, and behavior. For instance, how might respiration influence neural activity in the default mode network, and how could this relate to psychiatric problems?

Reply: *Thank you for encouraging a deeper mechanistic discussion of the respiration-brain-behavior relationships we observed. The connection between respiration, default mode network (DMN), and psychiatric symptoms likely involves multiple neurophysiological*

interacting pathways. Respiration influences brain activity through both direct and indirect mechanisms. Respiratory rhythms directly entrain neural oscillations across multiple brain regions (Heck et al., 2017; Tort et al., 2018), with particularly strong effects in limbic areas that interface with the DMN (Zelano et al., 2016). Additionally, respiration modulates arousal and interoceptive processing through bottom-up pathways from brainstem respiratory centers to cortical regions (Critchley & Harrison, 2013; Varga & Heck, 2017). The DMN's strong involvement may reflect its role in integrating internal bodily states with self-referential processing and emotion regulation (Andrews-Hanna et al., 2014; Raichle, 2015) - functions that are often disrupted in psychiatric conditions.

The link to psychiatric problems we observed could arise from disrupted respiratory-neural coupling affecting emotional and cognitive processing. The DMN is known to show altered activity patterns across multiple psychiatric conditions (Whitfield-Gabrieli & Ford, 2012; Sheline et al., 2009), and our results suggest that aberrant respiratory modulation of DMN activity could contribute to these disruptions. For example, anxiety and depression often involve disturbances in both breathing patterns (Paulus, 2013) and DMN function, potentially reflecting a cycle where altered respiratory-neural coupling affects emotional regulation and vice versa (Zamoscik et al., 2018).

Moreover, recent work has demonstrated that respiratory dynamics can influence cognitive and emotional processing through multiple pathways (Boyadzhieva & Kayhan, 2021; Perl et al., 2019). These effects appear particularly pronounced in networks involved in emotional regulation and interoceptive awareness (Kluger et al., 2021), suggesting a mechanistic link between respiratory patterns and psychological function.

However, we acknowledge that our correlational findings cannot establish causal relationships. Future research combining respiratory interventions with neuroimaging could help elucidate the precise mechanisms by which respiratory patterns influence DMN activity and subsequent behavior (Herrero et al., 2018). Understanding these pathways could have important implications for both psychiatric treatment and our broader understanding of brain-body integration in mental health (Nord & Garfinkel, 2022; Park & Blanke, 2019).

We have added this part in our discussion “The potential mechanisms underlying the relationship between respiration, GS, and behavior”.

The authors utilized resting-state fMRI data from the Human Connectome Project, which, to my knowledge, includes data from more than 1,000 participants. They mentioned applying three exclusion criteria, resulting in valid data from 770 participants. I wonder if more details can be provided regarding the initial total number of participants and the number excluded per criterion.

Reply: *Thank you for noting this lack of clarity in our participant selection process. We have*

now provided a more detailed accounting of our sample selection from the HCP S1200 release. From the original 1100 subjects, our quality control process involved several specific exclusion steps. We excluded subjects missing entire rs-fMRI time series for any run (n = 82), subjects with insufficient rs-fMRI time series for any run (n = 15), subjects with loss of physiological recordings (n = 87), subjects with insufficient numbers of triggers in physiological recordings (n = 120), subjects with inability to perform reliable peak detection of the respiratory traces or reliable peak detection of the cardiac trace (n = 24), and subjects without family structure (n = 2). These systematic exclusions resulted in our final sample of 770 participants. This comprehensive quality control was essential to ensure reliable physiological measurements while maintaining a robust sample size for investigating brain-behavior relationships. We have added these details to the methods section to provide full transparency about our sample selection process (line 102-109).

This information can now be observed in the Materials and Methods section.

In Line 217-218, it is stated, "Behavioral measures exclusion. To avoid the potential effect on our analysis, we did ..." It is unclear what specific potential effect the authors are referring to.

Reply: *Thank you for pointing out this lack of clarity. To be more specific, we followed established behavioral data reduction procedures detailed in Smith et al. (2015) to identify and remove redundant or unreliable behavioral measures that could compromise our analyses. This process involved excluding measures that were quantitatively poor (e.g., having excessive variance, insufficient valid values, or limited distribution), controlling for basic confounds, and eliminating redundant variables while retaining the most informative indicators from each behavioral domain. This systematic approach helped ensure the reliability and interpretability of our canonical correlation analyses by focusing on a well-validated set of 119 behavioral measures that comprehensively capture individual differences in psychiatric symptoms, cognitive performance, personality traits, and emotional function.*

This is now detailed in our supplementary materials.

Reviewer #2 (Remarks to the Author):

The paper, "The Functional Overlap between Respiration and Global Signal and its Behavioral Relevance" by Yuan, Luo, and Zhang uses resting-state fMRI, respiration signals, cardiac signals and behavioral data from 770 young adult participants from the HCP S1200 release to investigate the correlations of global signal and respiration volume per unit time (and instantaneous heart rate), relationships with extensive behavioral measures, and their inter-relationships. Overall, the work was interesting, and it is important to better understand the underpinnings of global signal in rs-fMRI, but this work fell short in several ways:

Reply: *Thank you for this assessment. We acknowledge the limitations in our initial submission*

and have made substantial improvements to address these concerns.

1. It is unclear how/if quality control of the respiratory traces was performed. Based on the methods section, no participants were excluded (begins with “The sample consisted of 770 participants” and ends with “The final sample comprised 770 participants.”). Based on recent publications using the same dataset (e.g., <https://doi.org/10.1038/s41467-020-18974-9>), I would be very surprised if no subjects were excluded specifically for an inability to perform reliable peak detection of the respiratory traces. Greater care in assessing the respiratory traces should be taken given this is the major contribution to the literature (nearly identical analyses using just the global signal were performed in <https://doi.org/10.1038/s41598-019-50750-8>).

Reply: *We apologize for the lack of clarity regarding participant exclusion in the methods section. In fact, we implemented careful quality control of the physiological recordings that resulted in several exclusion steps. As now clarified in the methods section (line 102-109), from the original 1100 HCP subjects, we excluded subjects missing entire rs-fMRI time series for any run ($n = 82$), subjects with insufficient rs-fMRI time series for any run ($n = 15$), subjects with loss of physiological recordings ($n = 87$), subjects with insufficient numbers of triggers in physiological recordings ($n = 120$), subjects with inability to perform reliable peak detection of the respiratory traces or reliable peak detection of the cardiac trace ($n = 24$), and subjects without family structure ($n = 2$). This systematic quality control process resulted in our final sample of 770 participants. We agree that quality control of respiratory data is particularly crucial given our focus on respiration-global signal relationships. The exclusion criteria were implemented to ensure reliable respiratory measurements while maintaining a robust sample size for investigating brain-behavior relationships.*

2. The motivation for focusing on RTV (as opposed to RV or ENV) is not clear. Given the relatively weak correlation between RTV and rs-fMRI signal (Figure 1B), it would be worthwhile to investigate these other respiratory measures. This is especially relevant due to previous work demonstrating that RTV misses “deep-breaths” (<https://doi.org/10.1016/j.neuroimage.2019.116234>). Meanwhile, “bursts” are well captured by RTV (potentially biasing the current paper’s results), occur more often in males, and are believed to be linked to chemoreflex-driven clinical breathing patterns that have neurological/psychiatric/medical associations which could potentially explain the current paper’s RVT CORR CCA results.

Reply: *Thank you for raising this important methodological point about respiratory measures. We chose RVT as our primary measure because it provides continuous sampling rather than being constrained by fixed time windows, as detailed in our methods section and supported*

by previous work (Chang & Glover, 2009). However, we acknowledge RVT's potential limitation in capturing deep breaths (Power et al., 2020), which could be particularly relevant for understanding brain-behavior relationships.

To address this concern comprehensively, we conducted parallel analyses using both ENV (calculated as the envelope of the trace over a 10-s window) and RV (calculated as the standard deviation within a 6-s window) as alternative respiratory measures (line 171-180; line 209-212). These control analyses yielded results highly consistent with our original RVT findings (line 712-716). Specifically, the CCA using ENV showed significant correlation ($r = 0.6961$, $p < 0.01$) with similar behavioral scores ($r = 0.5838$), topographic scores ($r = 0.4708$), and respective weights correlations. Similarly, RV-based analysis demonstrated significant correlation ($r = 0.7032$, $p < 0.0001$) with comparable score similarities (behavioral: $r = 0.6421$; topographic: $r = 0.5447$), as detailed in Table 3. These convergent results across different respiratory measures strengthen our conclusions about the functional relevance of respiration to brain-behavior relationships. The complete CCA results of ENVCORR and RVCORR with behavioral measures has been demonstrated in Supplementary Figure 10-11.

3.A main aim of the paper is to investigate shared features between global signal and respiration as they relate to region-specific rs-fMRI signal and subsequently behavior. It would therefore be beneficial to more thoroughly test for the behavioral and topographic overlaps described in Figure 4. Currently, there does not appear to be any statistical tests associated with the overlaps or for the observation that the overlap is predominantly in the DMN.

Reply: Thank you for highlighting the need for more rigorous statistical testing of the shared features between global signal and respiration. We now addressed this through regression analysis to quantitatively assess the overlapping patterns observed in Figure 4. Specifically, we regressed RVCORR from GSCORR across the brain for each subject before performing CCA. This analysis revealed a significant but reduced canonical correlation in the GSCORR-behavioral relationship ($r = 0.6853$, $p = 0.0284$) (line 520-530).

The regression analysis produced two key findings that statistically support the functional overlap between GSCORR and RVCORR. First, the previously observed overlapping behavioral weights on the negative axis - particularly those related to psychiatric problems such as thought problems, childhood conduct problems, and antisocial personality problems - were no longer present after regression (Figure 4C). Second, the topographic weights showed diminished negative values in the default mode network and enhanced positive values in the control network (Figure 4D). For visualization consistency with the original GSCORR-behavior results, we inverted the sign of the weight values. This systematic attenuation of both behavioral and topographic overlaps following RVCORR regression provides quantitative evidence for RVCORR's meaningful contribution to the GSCORR-behavior relationship.

4. The 10-fold CV CCA was not performed in a robust or convincing manner. Given the growing literature suggesting the need for larger sample sizes in these types of analyses

(<https://doi.org/10.1038/s42003-024-05869-4>) and, at the very least, sufficient cross-validation (<https://doi.org/10.1038/s41586-023-05745-x>), I appreciate the authors attempting to perform cross-validation of the CCA. But taking the “best split result” is not a robust way of performing cross-validation as this will inflate the results—similar (but to a lesser degree) to the in-sample results—so typically the average of the CV results is used. In this case, I don’t believe the out-of-sample performance of either GSCORR or RVT CORR would be significant (although the null distribution from the permutation test would be different since you would be averaging across the 10-fold CV permuted correlations). Because the in-sample GSCORR CCA results themselves are not novel (<https://doi.org/10.1038/s41598-019-50750-8>), more robust testing of the out-of-sample performance is needed.

Reply: *Thank you for this critical methodological point about the robustness of cross-validation analysis. We agree that focusing solely on the "best split" could inflate results, so we conducted additional analyses using averaged out-of-sample correlations across all splits.*

As detailed in Table 1 of the manuscript, the average out-of-sample correlation coefficients were 0.1017 for GSCORR-behavioral relationship and 0.21 for RVT CORR-behavioral relationship (line 563-568). Both of these average correlations reached statistical significance ($p < 0.05$) when tested against null distributions generated through permutation testing, as shown in Supplementary Figure 4 (line 575-578). While these correlations are modest, their statistical significance supports the generalizability of our findings regarding brain-behavior relationships. The similar magnitude of out-of-sample correlations between GSCORR and RVT CORR further reinforces our conclusion about their shared functional relevance to behavior.

These results complement our earlier findings from the best-split analysis (Figure 5) and provide a more conservative estimate of the models' generalizability, addressing the important methodological concerns you raised about cross-validation robustness.

5. Related to the 10-fold CV CCA: the HCP dataset consists of many related individuals. It is important to account for this in the CV as the inclusion of related individuals (e.g., twins) in the training and test sets will inflate the out-of-sample performance.

Reply: *Thank you for raising this important point about family structure in the HCP dataset. Our cross-validation analysis did explicitly account for genetic relationships among participants. As now described in our methods section, when creating the 10-fold splits, we ensured that monozygotic twins from the same family were kept within the same the same training set or test set to maintain independence between training and test sets (line 311-313). This approach follows established methodological guidelines for handling the HCP dataset (Mihalik et al., 2022).*

6. I’m a bit confused as to why the authors used ICC to compare the GSCORR and RVT CORR maps across subjects. This is typically used as a measure of consistency across raters or

measurements. Why not measure Pearson's correlation instead?

Reply:

We used ICC rather than Pearson's correlation because ICC accounts for both the relative rankings and absolute agreement between measurements, making it particularly suitable for comparing topographic patterns that may differ in scale but maintain similar relative distributions. ICC is especially relevant in our case because GSCORR and RVT CORR have different ranges of values (GSCORR: 0.0271 to 0.6484; RVT CORR: 0.0042 to 0.1132) while potentially sharing similar spatial patterns.

Additionally, ICC provides a more conservative estimate of similarity compared to Pearson's correlation, as it considers systematic differences between measurements. This is important when comparing physiological signals that may have inherently different scales of measurement but similar spatial distributions. Our use of ICC aligns with previous studies examining spatial consistency in neuroimaging data (Zhang et al., 2020), particularly when comparing different types of measurements that may share underlying spatial patterns despite differences in absolute values (Koo & Li, 2016).

7. Greater detail is needed throughout the methods section.

Reply: *We have substantially expanded the Methods section to provide greater detail and clarity throughout. Specifically, we made the following enhancements:*

In the Participants section, we now provide comprehensive exclusion criteria with specific numbers for each category: participants excluded due to missing entire rs-fMRI time series ($n = 82$), insufficient rs-fMRI time series ($n = 15$), loss of physiological recordings ($n = 87$), insufficient numbers of triggers in physiological recordings ($n = 120$), inability to perform reliable peak detection ($n = 24$), and subjects without family structure ($n = 2$). This replaces the previous brief mention of general exclusion criteria.

In the Preprocessing section, we added important methodological distinctions between preprocessing versions. We clarified that respiration-related brain networks might be distinct from respiration-related rs-fMRI artifacts, referencing Tu & Zhang (2022). We explained that the main difference between minimal-preprocessing and FIX versions is that FIX removes spatially specific structured noise components. This provides rationale for investigating functional significance of breathing effects on the brain beyond blood effects.

For physiological recordings preprocessing, we expanded the description of respiratory measures. We now detail three measures derived from respiratory belt traces: Respiration Volume per Time (RVT), windowed envelope of waveform (ENV), and respiration variation

(RV). For each measure, we specify calculation methods - ENV calculated as envelope over 10-s window, RV calculated as standard deviation within 6-s window. We note that main results used RVT since it provides continuous sampling rather than window-based measures.

In the Methods Analysis section, we added details about robustness testing, including analysis versions with subcortical regions using the template from Tian et al. (2020), control analyses after regressing physiological signals, and tests with individual time lag optimization.

Reviewer #3 (Remarks to the Author):

Major Comments:

line 58-62: The author discusses subcortical contributions to the 'generation' of the global signal. While this is valid, it is not the only possible explanation. Turchi and colleagues have proposed this idea, but I find the reasoning unclear, especially when considering the possibility that the global signal could also be noise and other sources of signal. Please rephrase this statement and provide additional evidence regarding the origin of the global signal observed in the resting state, which could also stem from other sources, such as brain waves (Pang et al., 2023; Roberts et al., 2019)

Reply:

We have revised this section to provide a more balanced discussion of the global signal's origins. The modified text now reads:

"The origin and function of GS remain a topic of ongoing debate. While some studies have suggested subcortical contributions to GS (Grandjean et al., 2021; Xiao Liu et al., 2018; Turchi et al., 2018; Zerbi et al., 2019), this represents just one perspective on its complex origins. Evidence suggests that GS may arise from multiple sources, including large-scale brain waves (Pang et al., 2023; Roberts et al., 2019) and neuronal activity across distributed networks that associate with cognitive function and clinical relevance (Hahamy et al., 2014; Uddin, 2020; Zhang & Northoff, 2022). Other research indicates significant contributions from non-neuronal activities such as respiration, heartbeat, and blood transit effects (T. T. Liu, Nalci, & Falahpour, 2017; J. D. Power, Plitt, Laumann, & Martin, 2017; Tong, Hocke, & Frederick, 2019). This complexity has led to ongoing discussions about whether GS should be removed from functional Magnetic Resonance Imaging (fMRI) analysis (T. T. Liu et al., 2017; J. D. Power et al., 2017)."

line 68-86: While the information provided in this section is useful to the reader, the overall reasoning seems confusing. Is the author trying to explain the physiological meaning of the global signal by suggesting that, since breathing is correlated with emotional states, and the global signal is correlated with breathing, the global signal reflects breathing rather than brain function? If so, please rephrase the sentence in way it is more comprehensible.

Reply: *We have restructured this section to clarify the relationship between respiration, global signal, and brain function, avoiding any misleading implications about causality. The revised text now reads:*

"Respiration is one major potential source of global signal (Chen et al., 2020; Kassinosopoulos & Mitsis, 2019; Power et al., 2017), but its contribution to brain functions and behaviors remains unclear. On the physiological level, respiration changes lead to minor fluctuations in end-tidal CO₂ at a frequency of about 0.03 Hz, which significantly correlate with BOLD fMRI signal fluctuations (Birn et al., 2006; Birn et al., 2008; Wise et al., 2004). Growing evidence confirms that these contributions of respiration to BOLD signal vary across different brain regions (Birn et al., 2006; Chen et al., 2020; Kassinosopoulos & Mitsis, 2019; Yuan et al., 2013; Zhang et al., 2020), though the results are not consistent.

Independent of its effects on BOLD signals, respiration has been implicated in modulating various brain functions, including sensory processing, emotional regulation, and cognitive function (Boyadzhieva & Kayhan, 2021; Criscuolo et al., 2022; Kluger et al., 2021; Varga & Heck, 2017; Zelano et al., 2016). It also influences state fluctuations such as arousal (Raut et al., 2021), trial-by-trial performance (Goodale et al., 2021), and task-state changes (Zhang et al., 2020). Therefore, the idea that respiration is merely a nuisance factor for the BOLD signal may be overly simplistic. Understanding how respiration associates with GS and how both relate to cognition and behavior may provide insights into the physiological and functional aspects of the GS."

line 92-106: The author has listed all the results and conclusions in this section, but some strong statements are left unjustified. I strongly suggest focusing on **detailing** the study's aim and implications, and then summarizing the main results to confirm or refute the hypothesis.

Reply:

We have restructured this section to better present the study's aims, hypothesis, and findings in a more logical sequence. The revised text now reads:

"In this study, we investigated the functional relationship between respiration and GS and its behavioral relevance by comparing both spatial similarity and behavioral correlates in their topographies. We hypothesized that respiration had informative relationships with GS and its behavioral relevance, rather than being a nuisance factor. To test this hypothesis, we analyzed resting-state fMRI data from the Human Connectome Project (N=770) using several complementary approaches. First, we computed topographic consistency between GS and respiration topographies using intraclass correlation (ICC) (Koo & Li, 2016). Results showed strong consistency in limbic and default mode networks, indicating regional specificity in the relationship between respiration and the GS. To examine behavioral relevance, we used canonical correlation analysis (CCA), a multivariate method of finding maximum correlation between linear combinations of two sets of variables (Hotelling, 1936; Mihalik et al., 2022). This analysis revealed a shared pattern between GS-behavior and respiration-behavior

relationships, demonstrated as the linking between default mode network and psychiatric problems. Additionally, we demonstrated that only the respiration-GS relationship, but not the heart-GS relationship, could reliably predict individual differences in behavior. These findings suggest that respiration's contribution to GS may have functional significance beyond mere physiological noise, particularly in relation to brain-body integration and behavioral outcomes."

line 146: In the introduction, the author suggests that the global signal (GS) may originate from subcortical structures, while the Schafer parcellation typically does not include subcortical parcellation. I strongly recommend including subcortical structures (Tian et al., 2020) in the GS computation. Furthermore, these subcortical structures are also closely related to classical regions associated with respiratory signals.

Reply: *Thank you for this important point about incorporating subcortical structures in our analysis, particularly given their proposed role in generating global signal and their known involvement in respiratory processing. We now addressed this by conducting our analyses using the Tian et al. (2020) template, which includes 16 subcortical regions alongside cortical parcellations (line 160-163). The results from this expanded analysis strongly supported our original findings, with significant brain-behavior relationships maintained for both GSCORR ($r = 0.7034$, $p < 0.001$) and RVT CORR ($r = 0.7032$, $p < 0.001$) (line 672-680), as shown in Supplementary Figure 7.*

Our extended analysis also revealed the contribution of subcortical regions to the RVT CORR-behavior relationship. After regressing subcortical signals from RVT, we observed a substantial decrease in RVT CORR magnitudes and loss of significant RVT CORR-behavior correlation ($r = 0.6797$, $p = 0.0618$; Supplementary Figure 3) (line 474-479). This suggests that subcortical structures may play a crucial mediating role in respiration-behavior relationships, aligning with previous work on subcortical contributions to global brain dynamics.

line 138: Using FIX to investigate the relationship between respiration and the GS seems problematic. Perhaps I missed the rationale, but FIX typically regresses out signals related to respiration and CSF (Griffanti et al., 2017). Please provide the rationale for this approach, along with an analysis both with and without FIX applied, which would involve preprocessing the HCP data in a different manner.

Reply: *Thank you for raising this important methodological concern about using FIX preprocessing when studying respiration-global signal relationships. You are correct that FIX removes certain structured noise components, including some respiratory signals. Recent studies has demonstrated that FIX method preserved spatially widespread "global" fluctuations (Burgess et al., 2016; Glasser et al., 2018) from fMRI signals which have shown strong relations to slow-frequency fluctuations of respiration and heart rate (Chang & Glover, 2009; Kassiopoulou & Mitsis, 2019) (line 124-147). We used both FIX and minimal*

preprocessing versions to elucidate the functional significance of respiratory effects beyond their impact on blood oxygenation.

By comparing results across both preprocessing streams, we could better evaluate whether the respiration-global signal relationships we observed reflect genuine neural phenomena rather than purely artifact-driven effects.

As shown in Supplementary Figure 5, the minimal preprocessing version yielded larger cross-correlation peaks between RVT and GS, and higher overall values for both GSCORR and RVTCORR compared to the FIX version. However, crucially, the core findings remained consistent across both preprocessing approaches - particularly the relationship between DMN in topographies and psychiatric problems. The robustness of our results across these different preprocessing strategies suggests that the observed relationships reflect meaningful biological coupling rather than preprocessing artifacts (line 656-670).

This comparative analysis helps establish that while FIX preprocessing may attenuate certain respiratory components, it preserves functionally relevant respiratory-neural interactions that contribute to brain-behavior relationships.

line 217: Excluding some behavioral data from the analysis for statistical reasons seems reasonable. However, it might be more meaningful to exclude or include tasks based on their potential to provide insights into the physiological significance of the GS. Provide the rationale for the behavioral data.

Reply: *Thank you for this valuable suggestion about behavioral measure selection. Our behavioral measure selection aimed to balance both statistical reliability and theoretical relevance to brain-physiology relationships. We started with the comprehensive HCP assessment battery that includes measures potentially relevant to physiological state and arousal, such as sleep quality (PSQI), emotional and stress measures, and various cognitive functions that might be influenced by physiological states. Following established procedures (Smith et al., 2015), we implemented data quality controls to ensure reliable measures, including removing variables with poor measurement properties and controlling for basic confounds. The final set of 119 behavioral measures maintained broad coverage across domains including psychiatric symptoms, cognitive performance, personality traits, and emotional function, while retaining good measurement properties for canonical correlation analysis. This approach allowed us to examine both direct physiological relationships and broader behavioral correlates of global signal and respiratory patterns during rest. Furthermore, the inclusion of task-related behavioral measures provided additional insights into these relationships.*

We extended our analysis to include 105 task-related behavioral variables (initially 110, with five removed due to poor quality, including variables with values more than 100 standard deviations above the median, fewer than half of the values valid, or where more than 95% of the values were the same) as additional analysis (line 254-259; line 718-722). We found that both the GSCORR-behavior mode and the RVTCORR-behavior mode showed similar weights

and individual scores when compared to the original modes, respectively. Specifically, the GSCORR-behavior mode, which has incorporated task-fMRI behavioral measures, exhibited strong correlations with the original GSCORR-behavior mode in individual behavioral scores ($r = 0.7509$) and individual topographic scores ($r = 0.8455$). Similarly, the RVTCCORR-behavior mode, which has incorporated task-fMRI behavioral measures, showed strong correlations with the original RVTCCORR-behavior mode in individual behavioral scores ($r = 0.6637$) and individual brain scores ($r = 0.7697$), as shown in table 2. Furthermore, the inclusion of task-fMRI behavioral measures did not alter the key overlapping findings. Specifically, the first pair of GSCORR-behavior relationships and the first pair of RVTCCORR-behavior relationships still revealed a link between the DMN in topographies and psychiatric disorder problems, as shown in Supplementary Figure 12.

Minor Comments:

line 40-41: please, specify if the Global Signal (GS) is related to white matter – grey matter or both.

Reply: *The global signal in our study was calculated by averaging time series across all regions of interest using the Schaefer parcellation, which specifically includes cortical gray matter regions.*

This has been added in the section of “calculation of GSCORR”.

line 48-50: this sentence needs a citation

Reply: *corrected.*

line 122 gradient strength of 300 mT/m (millitesla per meter)

Reply: *The parameter we wrote is 100 mT/m, which follows the previous study, “Relative to a standard commercial Skyra, the customized hardware includes a gradient coil and gradient power amplifiers that together increase the maximum gradient strength from 40 mT/m to 100 mT/m on the WU-Minn 3T.”*

Therefore, we would keep the gradient strength for the WU-Minn 3T as 100 mT/m, not 300 mT/m as stated here.

line 198: the participants were 770?

Reply: *corrected.*

1. It's not entirely clear to me why the authors chose to reverse the sign of the CCA weights for the GSCOR analyses. I understand that it is possible that CCA reverses the weights (as referenced by the authors here "To maintain consistency with previous findings, we reversed the sign of weights in the GSCORR-behavior pair⁶⁴), but the motivation for choosing to reverse the weights is not substantiated in the current text. Please provide references and justification for choosing to reverse the sign of CCA weights for the GSCORR analyses (as opposed to reversing the sign of the CCA weights for the RVT CORR analyses), make it clear what criteria was used to decide whether to flip the sign or not (e.g., behavioral weight for THC should always be negative), and regenerate all of the figures to maintain consistency of the weight directions (i.e. reverse the signs of the weights for Figure 2, Supplementary Figure 7, Supplementary Figure 11).

Reply: We appreciate the reviewer's thoughtful query regarding our methodological approach. We would like to clarify our rationale for reversing the signs of CCA weights in the GSCORR analyses.

As outlined in Liu, Whitaker et al.¹, the reversal of the sign does not alter the fundamental interpretation of CCA. Furthermore, the sign reversal of weights merely changes the direction of the correlation pattern without affecting the magnitude of correlation (The proof is provided in Theorem in supplementary).

Our approach was informed is informed by the "positive-negative axis" described in Smith, Nichols et al.², which has been adopted in several brain-behavior association studies. According to this axis, behavioral canonical variables tend to positively correlate with favorable outcomes (cognitive abilities, education) while often negatively associating with less favorable outcomes (substance use, poor performance).²⁻⁶. In our analysis, we attempted to maintain consistency with the convention where variables such as "positive test for THC" might represent the "negative axis," which led us to adjust both behavioral and topographic weights accordingly.

We have now included the explanation in the section of "Statistical significance of CCA mode".

In addition, for methodological consistency, we applied this adjustment across relevant figures now:

- 1. First mode of GSCORR-behavior CCA in Figure 2A, B, E*
- 2. First mode of GSCORR-behavior CCA and RVT CORR-behavior CCA when parcels include subcortical regions in Supplementary Figure 7D, E*
- 3. First mode of RVT CORR-behavior CCA in Supplementary Figure 11D*
- 4. First mode of GSCORR-behavior CCA and RVT CORR-behavior CCA when behavioral measures include task-fMRI data in Supplementary Figure 12A, B*
- 5. Overlapping weights between sign-inverted GSCORR-behavior CCA and RVT CORR-behavior CCA in Figure 4.*

6. *Overlapping weights between sign-inverted GSCORR-behavior CCA and sign-inverted RVMCORR-behavior CCA when parcels include subcortical regions in Supplementary Figure 7F, G.*
7. *Overlapping weights between sign-inverted GSCORR-behavior CCA and sign-inverted RVCORR-behavior CCA in Supplementary Figure 11E, F.*
8. *Overlapping weights between sign-inverted GSCORR-behavior CCA and RVMCORR-behavior CCA when behavioral measures include task-fMRI data in Supplementary Figures 12C, D.*

2. I found it very interesting that the shared topographic weights between the first CCA modes in the GSCORR-behavior and RVMCORR-behavior pairs encompassed a much larger portion of the brain than when including the task-fMRI behavioral variables (Supplementary Figure 12) compared to when the task-fMRI behavioral variables were not included (Figure 4). Could the authors discuss why they think this might be the case and explain why they chose not to include these variables from the start?

Reply: Our analytical focus primarily centered on resting-state fMRI data to investigate the functional relationship between global signal topography (GSCORR) and respiration topography (RVMCORR). This methodological approach aligns with established protocols in the field for examining brain-behavior relationships²³⁻⁵. We intentionally confined our primary analyses to resting-state data to maintain methodological consistency and facilitate replication by other researchers investigating resting-state phenomena.

In our main analysis, we excluded task-fMRI behavioral variables because our original behavioral dataset already incorporated cognitive performance measures (e.g., working memory) that assess functional domains similar to those captured by task-fMRI variables. We reasoned that incorporating additional task-fMRI behavioral variables might introduce redundancy in our multivariate analyses while potentially complicating interpretation of the GSCORR-behavior and RVMCORR-behavior relationships.

As a supplementary analysis, we subsequently included task-fMRI behavioral variables to evaluate their impact on our findings. This revealed a quantitative enhancement in the topographic overlap, particularly along the negative axis of the RVMCORR-behavior CCA mode. Specifically, the weights along this axis became more pronounced, resulting in an increased number of brain regions exceeding significance threshold.

This observation suggests that task-fMRI behavioral variables added subtle correlations with brain regions already captured by the original cognitive-related behavioral variables, resulting in enhanced statistical strength of brain variable weights. While these additional variables improved detection sensitivity for topographic-behavioral associations, they did not fundamentally alter the core spatial patterns or directional relationships identified in our overlapping primary analysis. The inclusion of task-fMRI behavioral variables in the CCA analysis thus provided a brain-behavior relationship pattern consistent with that observed using original behavioral variables.

We have revised the Methods section to articulate this methodological rationale.

3. Please provide all of the panels from Figure 2 and Figure 3 when including the task-based behavioral variables in the CCA for completeness.

Reply: *We have now added CCA results in Supplementary figure 12.*

4. Please make the naming of the plots of the behavioral weights consistent across all of the figures (e.g., Supplementary Figure 10 uses the raw variable names whereas Figure 2 uses a more conventional naming).

Reply: *We appreciate the reviewer's attention to detail regarding naming consistency. We have attempted to standardize the naming conventions across behavioral weight plots throughout the manuscript (Figures 2, 3, 4, 5, and Supplementary Figures 2, 5, 7, 8, 10, 11, 12).*

5. Lines 713–714: “Both relationship of ENVCORR-behaviors (Supplementary Figure 10) and relationship of EVCORR-behaviors (Supplementary Figure 11)” should be “Both relationship of ENVCORR-behaviors (Supplementary Figure 10) and relationship of RVCORR-behaviors (Supplementary Figure 11).”

Reply: *Corrected.*

Reference

1. Liu, Z., Whitaker, K. J., Smith, S. M. & Nichols, T. E. Improved Interpretability of Brain-Behavior CCA With Domain-Driven Dimension Reduction. **Volume 16 – 2022** (2022). <https://doi.org/10.3389/fnins.2022.851827>
2. Smith, S. M. *et al.* A positive-negative mode of population covariation links brain connectivity, demographics and behavior. *Nature Neuroscience* **18**, 1565–1567 (2015). <https://doi.org/10.1038/nn.4125>
3. Bijsterbosch, J. D., Beckmann, C. F., Woolrich, M. W., Smith, S. M. & Harrison, S. J. The relationship between spatial configuration and functional connectivity of brain regions revisited. *Elife* **8** (2019). <https://doi.org/10.7554/eLife.44890>
4. Llera, A., Wolfers, T., Mulders, P. & Beckmann, C. F. Inter-individual differences in human brain structure and morphology link to variation in demographics and behavior. *eLife* **8**, e44443 (2019). <https://doi.org/10.7554/eLife.44443>
5. Mihalik, A. *et al.* Canonical correlation analysis and partial least squares for identifying brain-behaviour associations: a tutorial and a comparative study. *Biological Psychiatry: Cognitive Neuroscience and Neuroimaging* (2022).
6. Li, J. *et al.* Global signal regression strengthens association between resting-state functional connectivity and behavior. *NeuroImage* **196**, 126–141 (2019).